# Exosomal Biomarkers: A Comprehensive Overview of Diagnostic and Prognostic Applications in Malignant and Non-Malignant Disorders

**DOI:** 10.3390/biom15040587

**Published:** 2025-04-15

**Authors:** Mahda Delshad, Mohammad-Javad Sanaei, Mohammad Hossein Mohammadi, Amir Sadeghi, Davood Bashash

**Affiliations:** 1Department of Hematology and Blood Banking, School of Allied Medical Sciences, Shahid Beheshti University of Medical Sciences, Tehran 1985717443, Iran; delshad.mahda@gmail.com (M.D.); javadsanaei137@gmail.com (M.-J.S.); drmohammadi@sbmu.ac.ir (M.H.M.); 2Department of Laboratory Sciences, School of Allied Medical Sciences, Zanjan University of Medical Sciences, Zanjan 1411718541, Iran; 3Gastroenterology and Liver Diseases Research Center, Research Institute for Gastroenterology and Liver Diseases, Shahid Beheshti University of Medical Sciences, Tehran 1985717411, Iran; amirsadeghimd@yahoo.com

**Keywords:** exosome, diagnosis, prognosis, clinical trial, malignancy disorders, non-malignancy disorders

## Abstract

Exosomes are small extracellular vesicles, ranging from 30 to 150 nm, that are essential in cell biology, mediating intercellular communication and serving as biomarkers due to their origin from cells. Exosomes as biomarkers for diagnosing various illnesses have gained significant investigation due to the high cost and invasive nature of current diagnostic procedures. Exosomes have a clear advantage in the diagnosis of diseases because they include certain signals that are indicative of the genetic and proteomic profile of the ailment. This feature gives them the potential to be useful liquid biopsies for real-time, noninvasive monitoring, enabling early cancer identification for the creation of individualized treatment plans. According to our analysis, the trend toward utilizing exosomes as diagnostic and prognostic tools has raised since 2012. In this regard, the proportion of malignant indications is higher compared with non-malignant ones. To be precise, exosomes have been used the most in gastrointestinal, thoracic, and urogenital cancers, along with cardiovascular, diabetic, breathing, infectious, and brain disorders. To the best of our knowledge, this is the first research to examine all registered clinical trials that look at exosomes as a diagnostic and prognostic biomarker.

## 1. Introduction

Both prokaryotic and eukaryotic cells produce exosomes, which are small (30–150 nm), phospholipid bilayer extracellular vesicles used for signaling and intercellular communication [1]. Exosome is a biogenesis-related term exhibiting origin from the endosomal system. Being typically smaller than 200 nm, exosomes are a subtype of small extracellular vesicles (EVs) [2]. Indeed, they were formerly thought to be extracellular vesicles that expel undesirable cellular trash. Nevertheless, several studies have shown that exosomes are crucial molecular mediators for interaction between cells in the movement of different nucleic acids, proteins, and metabolites within the body [3,4]. Numerous cell types, such as immune cells [5], cancer cells [5], and stem cells [6], also produce exosomes. Exosomes are essential for immunological response, illness development, physiological control, and disease progression because of their many roles in information transmission between cells [7]. In practice, they have been considered the best options for applications pertaining to early illness diagnosis and biomarker creation [8].

Exosomes are released into the bloodstream as nanoparticles. Therefore, a wide variety of exosomal components may be used for illness monitoring, diagnosis, and prognosis. When diagnosing cancer, the content of exosomes, including proteins, miRNA, and lncRNA, as well as their number, are crucial [9,10]. Circulating exosomes have higher levels of exosomal markers (CD63, CD81, and CD9) [11,12], a few tumor antigens (carcinoembryonic antigen [CEA], CA125) [13], and stemness markers (epithelial cell adhesion molecule [EpCAM], CD24) [14], all of which may be useful for diagnosis. Exosomal programmed death-ligand 1 (PD-L1), for example, has been shown in clinical settings to be a possible predictor of response to anti-PD-1 treatment in patients with non-small cell lung cancer (NSCLC) and melanoma [15].

A sequence of imaging scans is used to diagnose different tumor forms, and a biopsy is then performed to verify the diagnosis [16]. In addition to being intrusive, these techniques may be expensive and painful for the patient. Additionally, individuals are more probable to acquire a late-stage diagnosis if regular screening is not performed. One possible disadvantage of a delayed diagnosis is that it may reduce the patient’s reaction to treatment and, therefore, their chances of survival [17]. Because of this, it is critical to track how well a patient is responding to treatment. Many traditional cancer detection methods are inappropriate for patients because they need intrusive procedures, such as surgeries, to collect tissue samples for examination. These operations may be risky and uncomfortable [18]. Furthermore, they may be expensive, burdening patients and healthcare systems alike, especially when many tests are needed to verify a diagnosis or track the effectiveness of therapy. As a consequence, smaller tumors or malignant cells may go undetected until they have progressed to more severe and untreatable phases [19,20]. A growing effort has been made to develop less invasive, cost-effective, highly sensitive diagnostic techniques, including liquid biopsies and sophisticated imaging technologies, that have the potential to detect tumors early and monitor patients undergoing therapy in order to conquer these limitations [21]. By providing earlier detection, less invasiveness, and more accurate monitoring, these advancements have the potential to completely transform the diagnosis of cancer and strengthen our fight against this terrible illness.

In the area of cancer detection, exosomes have become a rapidly developing research area with enormous promise [22]. Exosomes are also essential for a number of pathological as well as physiological processes, such as immunological responses, cancer, pregnancy issues, and cardiovascular problems [23]. Exosomes were discovered to be present in almost every bodily fluid, including blood [24], urine [25], cerebrospinal fluid (CSF) [26], saliva [27], pleural effusion [28], ascites fluid [29], amniotic fluid [30], breast milk [31], and bronchoalveolar lavage fluid (BALF) [32], according to the many investigations that followed. The pilot study evaluated saliva collection methods for exosome isolation and characterized exosomal proteins in cancer-free controls, oral potentially malignant disease (OPMD), and oral squamous cell carcinoma (OSCC) patients. A three-protein panel (AMER3, LOXL2, and AL9A1) distinguished cancer-free individuals from OPMD/OSCC with high accuracy (AUC 0.93). Another panel (PSB7, AMER3, and LOXL2) showed strong diagnostic potential for OSCC (AUC 0.96), suggesting its promise as a biomarker for early detection [33]. Another study found significant differences in the size, concentration, and biomolecular composition of salivary exosomes between healthy individuals, tobacco consumers, and oral cancer patients. Tobacco users and cancer patients had larger and more concentrated exosomes, suggesting early cellular changes. These findings highlight the potential of salivary exosomes as noninvasive biomarkers for distinguishing high-risk individuals and enabling early oral cancer detection [34]. Moreover, exosomal plasmatic levels before surgery could be a reliable indicator of survival and early recurrence in OSCC. It was shown that identifying peripheral exosomes can be a new clinical treatment technique for OSCC that may have consequences for prognostic evaluation [35]. Sweat is a simple and less interference-prone biofluid, making it valuable for disease biomarker studies. Sweat exosome-based detection shows potential for lung cancer screening through molecular expression analysis [36]. Notably, Dermcidin (DCD) and prolactin inducible protein (PIP) serve as biomarkers, with DCD linked to breast cancer and lymph nodes, while PIP is overexpressed in breast and prostate cancer [37]. Tear exosomes contain higher levels of exosome markers (CD9 and CD63) than serum exosomes and were analyzed using qRT-PCR and Western blot. Breast cancer-specific miR-21 and miR-200c were highly expressed in tear exosomes from metastatic breast cancer patients compared to healthy individuals [38]. Cholangiocarcinoma (CCA) is an aggressive cancer with poor prognosis and limited diagnostic or treatment options. This study identified two exosomal lncRNAs, ENST00000588480.1 and ENST00000517758.1, with significantly increased expression in bile from CCA patients. Their combined diagnostic performance (AUC: 0.709, sensitivity: 82.9%, specificity: 58.9%) correlated with cancer progression and poor survival. KEGG analysis highlighted their involvement in the p53 signaling pathway, suggesting their potential as noninvasive bile-based biomarkers and therapeutic targets for CCA [39]. For the first time, authors provided proof that tumor acidity and exosome the amount represent common cancer phenotypes. The results of an intriguing study revealed that pH 6.5 caused an astonishing rise in exosome release and buffering the medium drastically decreased the exosome release in all cancers [40].

Exosomes may be useful biomarkers for quick diagnosis and illness monitoring, according to some research [41]. Conventional tissue biopsy may not be able to accurately depict tumor heterogeneity and start treatment on time. On the other hand, liquid biopsy, which includes that of exosomes, circulating tumor cells (CTCs), and cell-free DNA (cfDNA), offers comprehensive information on tumors in a noninvasive way. A valuable source of cancer biomarkers is liquid biopsies. Compared to conventional serological markers, exosomal proteins have special characteristics. Exosomal proteins, for example, are more sensitive than proteins that are directly found in blood [42]. Moreover, compared to secretory proteins, exosomal proteins exhibit greater specificity [9]. Finding a single diagnostic biomarker is probably uncommon since tumors are so diverse. Exosomes may function as a mix of panel candidate indicators, such as RNA, mutant DNA, and oncogenic proteins.

Herein, we mainly introduce the advantages of exosomes as a liquid biopsy and focus mainly on exosomes, with particular emphasis on their composition and their application as a potential diagnostic marker in some common malignant tumors, pregnancy disorders, cardiovascular diseases, organ transplantation, and other disorders. Finally, we provide an overview of exosomes currently under review in clinical trials.

## 2. Physiological Functions of Exosomes

### 2.1. Cell-to-Cell Communication

Exosomes are essential in the complex network of intercellular communication inside biological systems. These nanoscale vesicles act as crucial messengers between cells, facilitating the intercellular transfer of various molecular cargo, including proteins, nucleic acids (DNA and RNA), and lipids [43,44]. Exosomes, secreted by various cell types, including immune, neural, and stem cells, are likely implicated in numerous physiological processes, such as antigen presentation [45], RNA transfer [46], and tissue repair [47]. Immune cells utilize exosomes to transmit signals that coordinate immune responses, whereas cancer cells exploit exosomal communication to facilitate tumor growth and metastasis [48,49].

It was shown that a novel function of exosomes in cell–cell communication related to atherosclerotic prevention [50]. Smooth muscle cells (SMCs) are pivotal in atherosclerosis, characterised by plaque formation within an artery, resulting in myocardial infarction. The research indicated that endothelial and smooth muscle cells interact through exosomes [50].

### 2.2. Waste Removal

Exosomes are vital tools for the elimination of waste and the mechanisms that control the quality of cellular metabolites within the network that is responsible for the maintenance of cellular metabolites [51]. Cells can make use of the remarkable ability of exosomes to enclose and transport a wide range of undesirable or degraded cellular components, such as misfolded proteins, broken organelles, and waste of metabolites, to facilitate the efficient elimination of these components from the cell [52,53] to facilitating their removal from the cell. In addition, the function of exosomes in the elimination of waste goes beyond the confines of the individual cell. Exosomes have the ability to be discharged into extracellular environments, such as physiological fluids, where they can contribute to the elimination of waste throughout the body and possibly assume a part in the process of intercellular signaling [54,55].

### 2.3. Development and Tissue Repair

Exosomes coordinate the mechanisms of development and tissue repair by enabling the delivery of growth factors and other signaling molecules to specific target cells [56,57]. In tissue regeneration, exosomes function as specialized messengers, directing and enhancing cellular responses essential for the repair and reconstruction of damaged or injured tissues. Exosomes are essential for neuronal communication and maintenance in the nervous system [58]. They function as nanocarriers, transporting neurotransmitters and various signaling substances between neurons and other supporting cells in the brain. This intercellular exchange underlies essential processes in neural development and aids in the refinement of neural circuits, emphasizing the significant influence of exosomes on neurological function and their potential therapeutic applications in neurodegenerative diseases and neurological disorders [59,60].

### 2.4. Immune Response Regulation

Exosomes play a vital role in coordinating immune responses, exerting their impact within the complex realm of immune system functionality [61]. Immune cells utilize exosomes to distribute signaling molecules that regulate immune responses, thereby regulating the intricate interactions of immunological systems [62,63,64]. Exosome release is essential to sustain robust immunostimulatory interactions between mature dendritic cells and B lymphocytes, which tightly associate with follicular dendritic cells to present antigen-MHC-II complexes to T lymphocytes. Moreover, the consequences of immunological activation may be facilitated by exosome-induced proliferation and survival of haematopoietic stem cells, as well as the activation of natural killer cells [65]. It has been shown that exosomes produced from dendritic cells overexpressing IL-4 or IL-10 inhibited delayed-type hypersensitivity reactions in an MHC-II-dependent manner in a murine model. Exosomes assume heightened importance as prospective diagnostic instruments in the context of disease. Cancer cells utilize exosomes as vehicles for disease-specific cargo, encompassing tumor-specific proteins and genetic material that further influence the immune response [66,67].

## 3. Diagnostic and Prognostic Function of Exosomes

For a number of different reasons, exosomes are an appealing target for application in clinical diagnostics and the development of biomarkers. Initially, the contents of exosomes, which include nucleic acids, proteins, and lipids, are altered when illness circumstances are present. Moreover, exosomes can be extracted from biological fluids that are easily accessible, such as urine, blood, and saliva, without the need for intrusive procedures. Early disease diagnosis is of utmost significance in the case of conditions affecting the central nervous system (CNS), and noninvasive availability makes this possible. The contents of exosomes are shielded from degradation since they are contained within a membrane-bound structure. This provides an advantage over typical specimens since the possible biomarkers are shielded from deterioration. Exosomes have a slower rate of degradation and are extremely stable in stored patient samples. Indeed, as exosomes are membranous structures, the contents could be adequately protected from degradation by extracellular proteases and can be highly stable in storage conditions [68]. This makes the application of exosomes in clinical settings possible because samples may be stored for extended periods of time before being analyzed. Due to the fact that exosomes exhibit surface markers that are associated with their cellular origin, it is possible to trace them back to their point of origin [69]. Nonetheless, the stability of exosomes could be influenced by storing conditions. In this regard, the biodistribution and uptake efficiency could be notably reduced following storage at 4 °C and −20 °C [70]. Exosomes, however, maintain their biodistribution at −80 °C for 14 days [71]. Also, it was shown that the number of exosomes, their size, their capacity to express protein at the surface, and functional stability could be constant at −70 °C after 25 days [72].

### 3.1. Diagnostic and Prognostic Function of Exosome in Non-Malignant Disorders

#### 3.1.1. Diagnostic Function of Exosomes in Cardiovascular Disease

Coronary artery disease (CAD), another name for heart disease, is still the world’s largest cause of mortality. In terms of human health in general, cardiovascular diseases (CVDs) are one of the biggest issues [73]. Circulating biomarkers of CVDs, like total cholesterol and low-density lipoproteins (LDL), and prognostic biomarkers for myocardial infarction (MI), like creatine kinase MB, high-sensitivity cardiac troponin, and high-sensitivity C-reactive protein, can only give a ballpark estimate of the likelihood that the disease will develop and progress [74]. In this sense, the novel blood-based exosome biopsy technique may provide a viable platform that might help with more accurate clinical diagnosis and prediction.

##### Exosomes in Coronary Artery Disease

The ability of exosomes from cardiosphere-derived cells to suppress apoptosis and increase angiogenesis was also demonstrated by miR-146a-bound exosomes, indicating the therapeutic usefulness of exosomes [75]. Exosomes carrying miR-210, miR-132, miR-181, miR-378b, miR-623, miR-941 (linked to ejection fraction enhancement), miR-1256, miR-384 (linked to fibrosis mitigation), miR-525-3p, miR-515-5p, miR-1224 (linked to angiogenesis stimulation), and GATA4-responsive-*miR-451*, originating from cardiac progenitor cells, have identical cardioprotective properties [76]. Recently, Liu et al. [77] advanced the understanding of the therapeutic function of circulating endothelial cell-derived microvesicle miRNAs, specifically miR-92a-3p, in modulating the phenotypes of endothelial cells and vascular smooth muscle cells in atherosclerotic environments, potentially serving as a prognostic indicator for CAD. Conversely, miR-939-5p was downregulated in serum-derived exosomes from patients with myocardial infarction and suppressed angiogenesis through the nitric oxide signaling pathway [78]. Utilizing liquid chromatography coupled with tandem mass spectrometry (LC-MS/MS), Cheow et al. identified 252 upregulated extracellular vesicle proteins following myocardial infarction and proposed a potential panel for early myocardial infarction diagnosis, comprising apolipoprotein C-III, apolipoprotein D, platelet glycoprotein Ib alpha chain, complement C1q subcomponent subunit A, and complement C5 [79]. Acute coronary syndrome (ACS) has also been linked to a variety of exosomal miRNAs. More specifically, ACS patients with higher levels of miRNA-208a had a worse survival rate during a one-year follow-up period, and miRNA-208a expression is significantly higher in the serum exosomes of ACS participants compared to healthy controls. Serum exosomal miRNA-208a expression is significantly correlated with peak troponin levels, LDL, and Killip class in the ACS cohort [80].

##### Exosomes in Heart Failure

Overexpression of microRNAs miR-22, miR-320a, miR-423-5p, and miR-92b in serum and serum exosomes has been associated with heart failure (HF). For the purpose of diagnosing and determining the prognosis of systolic heart failure, these biomarkers can be utilized as specific indicators [81,82,83]. Patients with HF had higher levels of serum-based exosomes encoding p53-responsive microRNAs since the beginning of acute myocardial infarction. These miRNAs included miR-34a, miR-192, and miR-194. Additionally, it was observed that the expression levels of exosomes miR-194 and miR-34a, but not miR-192, were strongly connected with left ventricular diastolic size and left ventricular ejection percentage, when tested about one year following the beginning of acute myocardial infarction (AMI) [84]. Exosome-associated p53-responsive microRNAs have the potential to predict left ventricular remodeling after the convalescent stage of AMI, as indicated by this research finding [81].

#### 3.1.2. Diagnostic Function of Exosomes in Lung Disorders

##### Asthma

Opportunities biomarkers for the diagnosis of asthma are exosomal miRNAs. These miRNAs have different expression patterns in asthmatic patients than in healthy people, and their levels are correlated with the phenotype and severity of the condition [85,86]. To distinguish individuals with moderate non-symptomatic asthma from healthy populations, for example, the exosomal miRNAs from the BALF of asthmatic patients vary significantly from those of healthy control participants [87]. Regarding the low type-2 asthma linked to obesity, it has been shown that plasma exosomal miRNA signatures contribute to lung function decline, which forms the foundation for a stratified treatment response [88]. In addition, there are other phenotypes of asthma, such as severe asthma, non-allergic asthma, and allergic asthma [89]. Differentiating between distinct phenotypes with the use of exosomal miRNA profiles enables more individualized treatment strategies. For instance, whilst lower levels of miR-21-5p, miR-126-3p, and miR-146a-5p may indicate neutrophilic asthma, higher exosomal levels of these two markers are suggestive of T2-high atopic asthma [90].

##### Pulmonary Fibrosis

As pulmonologists, pathologists, and radiologists work together in a multidisciplinary manner to diagnose pulmonary fibrosis (PF), it is still acknowledged that an accurate and timely diagnosis of PF is necessary. Fibrotic interstitial lung illnesses, like idiopathic pulmonary fibrosis (IPF), have a dismal prognosis [91]; therefore, early diagnosis is crucial. Exosomes found in bodily fluids, including blood and sputum, have been linked to PF, and studies have thus far demonstrated that they may serve as predictive, prognostic, and diagnostic indicators. Makon-Sébastien Njock’s research team identified IPF exosomal biomarkers in sputum specimens. Results showed downregulated miR-142-3p, miR-33a-5p, and let-7d-5p as promising diagnostic biomarkers. Negative correlations were found with diffusing capacity/alveolar volume of carbon monoxide [92]. Exosomal miR-142-3p’s potential as an IPF sputum biomarker was further supported by the discovery that it was markedly increased in the plasma of people with IPF [93]. This discovery reinforces the connection between IPF and exosomal miR-142-3p. By comparing IPF patients with healthy nonsmokers, Kaur et al. found 55 exosomal miRNAs that were differentially expressed in IPF human lung tissue. Additionally, they discovered that exosomes miR-22-3p, miR-320a-3p, miR-320b, and miR-24-3p were significantly upregulated in BALF-derived exosomes from IPF patients, whereas exosomes miR-375-3p, miR-200a-3p, miR-200b-3p, and miR-141-3p were significantly downregulated.

#### 3.1.3. Diagnostic Function of Exosomes in Liver Disorders

##### Acute Liver Injury

The quantity of liver-specific proteins in the extracellular vesicle (EVs) rose after hepatotoxin-induced liver damage, namely carboxylesterase (CES1), alcohol dehydrogenase 1 (ADH1), glutathione S-transferases (GST), apolipoprotein A1 (APOA1), albumin (ALB), haptoglobin (HP), and fibrinogen (FGB). It was shown that exosomal miR-122a-5p outperformed its serum equivalent in two animal models of acute liver injury (ALI) in terms of diagnostic performance, having a larger diagnostic time frame and an earlier diagnostic potential [94]. Furthermore, it has been shown that acetaminophen (APAP)-induced liver damage increased the amounts of liver-specific miRNAs in circulating exosomes, including miR-122, miR-192, and miR-155. However, after receiving therapy via the antioxidant N acetyl-cysteine (NAC), these levels dramatically dropped and returned to baseline, suggesting that exosomal miR-122, miR-192, and miR-155 levels reflected the degree of hepatocyte damage and could be employed as accurate diagnostic biomarkers for liver injury. Because they release large amounts of miRNAs that circulate in a tissue-specific manner, some cells are useful biomarkers for specific types of tissue damage. In this context, nine miRNAs were identified as ALI signatures. Of these, five miRNAs (miR-21a-5p, miR-92a-3p, miR-194-5p, miR-17-5p, and miR-19b-3p) were increased, whereas four (miR-451a, miR-27a-3p, miR-26a-5p, and miR-223-3p) were decreased [95]. Expression analysis of exosomal, serum, and hepatic miRNAs revealed the presence of many exosome-derived miRNAs, including miR-122a-5p. These miRNAs might be helpful in detecting acute liver damage caused by drugs like thioacetamide and paracetamol [94].

##### Non-Alcoholic Fatty Liver Disease and Alcohol-Associated Liver Disease

According to studies, exosomes may be useful diagnostic indicators for the course and severity of non-alcoholic fatty liver disease (NAFLD) because of their contents, which include microRNAs [96]. Zhang and Pan discovered 2588 miRNAs while monitoring blood exosomal microRNAs in children with non-alcoholic fatty liver disease. Children with NAFLD had different expression levels of 80 miRNAs than the control group, including the important miR-122-5p, miR-335-5p, and miR-27a [97]. Furthermore, a number of investigations have demonstrated that exosome contents, such as protein FZD-7, can vary and serve as predictive and diagnostic biomarkers for NAFLD [98,99].

Alcohol-associated liver disease (ALD)-affected mice had higher levels of circulating miR-155, and circulating exosomes were enriched in both miR-155 and miR-122 [100]. Subsequent examination of the miRNA content of exosomes showed that, in comparison to normal controls, mice with ALD and, more significantly, people with alcoholic hepatitis also had very high levels of miR-192 and miR-30a in their exosomes. It has been demonstrated that specific lncRNAs are expressed in the liver and serum of alcoholic cirrhosis patients [101]. The two most often expressed increased lncRNAs among them in persons with alcoholic cirrhosis were AK128652 and AK054921. Additionally, in individuals with alcoholic cirrhosis, AK128652 and AK054921 demonstrated an unfavorable relationship with survival [101] in 480 prospectively monitored patients. It is anticipated that these newly developed diagnostic and prognostic biomarkers will be prospectively investigated in bigger patient groups. Proteomic study of circulating EVs in a mouse model of ALD demonstrated a distinct cluster of EV-associated proteins that were elevated in comparison to control EVs [102]. One of these, heat-shock protein-90 (hsp90), was linked to a biological effect in macrophages following exosome transfer in both vitro and vivo. This suggests that EVs may have a role in cell-to-cell communication in addition to being a potential biomarker for EV-associated proteins [102].

##### Viral Hepatitis

Recent research has shown that by controlling (hepatitis B virus) HBV replication and transmission, exosomes can affect and contribute to HBV replication, transmission, diagnosis, and treatment. Immune-related miRNAs may play a part in inflammatory and immunological responses, according to prior research [103], Zhang et al. claim that by reducing the production of hepatitis B surface antigens (HBsAg), miR-199a-3p and miR-210 effectively reduced HBV replication. The prognosis of HBV infection is significantly influenced by exosomes. Zhao et al. confirmed the existence of proteins linked to liver cancer by analyzing the protein composition of exosomes from HBV-infected Huh7 cells and the control group. According to their research, several proteins in serum exosomes could be used as indicators of HBV and liver cancer linked to HBV [104]. miR-92a-3p and miR-146a-5p-containing serum exosomes have been identified as potential markers for assessing hepatic fibrosis in chronic hepatitis B. These indicators may be used to monitor the progression of liver fibrosis (LF) and distinguish between the disease’s early and severe stages [105]. There is a positive connection between exosomal miRNA-155 expression and hepatitis C virus (HCV) replication, as shown by the finding that exosomal miRNA-155 expression levels were similar to exosomal HCV RNA loads. Furthermore, the group infected with genotype respiratory syncytial virus 1b (SVR) had significantly higher exosomal miRNA-122 expression than those infected with genotype 2a and genotype 6a, indicating that the elevated exosomal expression of miRNA-122 may be associated with a positive therapeutic effect [106].

##### Liver Fibrosis

A better prognosis and efficient disease management depend on an early and accurate diagnosis of LF. Blood, serum, urine, saliva, amniotic fluid, and other biological fluids are among the fluids that contain exosomes [107]. Body fluid exosomes could be utilized to track the development of illnesses [3]. Serum exosomal miR-122 has been shown to be a viable diagnostic marker for LF in chronic liver illnesses with non-viral aetiologies. Serum exosomal miR-122 levels have been found to dramatically decline as LF progresses, potentially as a result of miR-122’s suppression, which encourages hematopoietic stem cell (HSC) proliferation and the expression of markers linked to fibrosis [108].

Exosomal miR-27a, a factor in the development of LF, is markedly raised in serum and strongly correlated with the severity of LF in metabolically associated fatty liver disease (MAFLD). Serum exosomal miR-27a has thus been proposed as a potential biomarker for the diagnosis of LF associated with MAFLD. Findings indicating an elevated expression of exosomal miR-574-5p in LF and a favorable correlation between it with collagen deposition and HSC activation point to the possibility of using it as a biomarker for the diagnosis of LF [109].

#### 3.1.4. Diagnostic Function of Exosomes in Pancreatitis

##### Acute Pancreatitis

There are two distinct exosome populations that are created in acute pancreatitis (AP), according to the findings of other investigations. These populations appear to differ from one another in terms of their origin, tissue distribution, molecular content, and physiological effects. When compared to pancreatitis-associated ascitic fluid (PAAF) exosomes, pancreatitis plasma exosomes have a higher concentration of inflammatory miR-155 and a slightly lower expression of miR-21 and miR-122 [110]. In addition, the rate of pro-inflammatory activity that plasma exosomes exhibit in macrophages is significantly higher than that of PAAF exosomes. A prospective investigation with a larger number of samples is now underway, and the plasma miRNA expression signature in AP patients has been established [111]. A study exhibited that 30 exosomal microRNAs were discovered to be elevated in AP. A further finding of the study was that pancreatic acinar cells are responsible for the regulation of macrophage activation through the secretion of exosomes that contain microRNAs. Furthermore, it was shown that the target genes of microRNAs that were differently expressed were responsible for regulating the activation of macrophages through the pathway of TRAF6-TAB2- TAK1-NIK/IKK-NF-κB [112].

##### Chronic Pancreatitis

At the time of an early clinical diagnosis, it might be challenging to differentiate between chronic pancreatitis (CP) and pancreatic ductal adenocarcinoma (PDAC). Exosomal miRNAs have demonstrated in multiple trials to be able to discriminate between patients with PDAC and those with CP. Exosomal miR-let7a expression is low in PDAC patients, but exosomal miR-10b, miR-20a, miR-21, miR-30c, miR-106b, and miR-181a are highly expressed in CP patients [113]. Additionally, after resection, exosomal miR-10b, miR-20a, miR-21, miR-30c, and miR-106b all decreased to normal levels [113]. Nakamura et al. used exosomal miRNAs in pancreatic juice to distinguish between individuals with CP and PDAC. Quantitative real-time reverse transcription polymerase chain reaction, or qRT-PCR, revealed that the PDAC patients expressed much more exosomal miR-21 and miR-155 than the CP patients. However, there were no statistically significant differences between PDAC and CP patients in terms of free miR-21 and free miR-155 expression levels [114].

##### Diabetes

Exosomes can be utilized as biomarkers for metabolic illnesses, according to recent research, allowing for the detection of disease risk signs and the potential for therapy or prevention. Consequently, it is critical to identify those who are at risk for diabetes and to encourage an early identification of the disease. Research that included cross-sectional and longitudinal groups of individuals with standard blood glucose and those with prediabetes or diabetes found that circulating exosome levels were considerably greater in patients with type 2 diabetes than in persons with normal blood glucose [115]. Additional research has examined the quantities of aquaporin 5 (AQP5) and AQP2 exosome-expelled in urine in 35 diabetic patients, indicating that exosomal AQP5 and AQP2 could serve as potential noninvasive biomarkers for the classification of diabetic nephropathy (DN) clinical stages [116]. Urinary exosomes may one day be employed, along with miRNA found in them, to aid in the early identification of many illnesses. One study evaluated the expression of miRNA in the urine exosomes of individuals suffering from type 1 diabetes mellitus who had early DN and those who did not. The results showed that urine exosomes from patients with microalbuminuria had a smaller amount of miR-155 and miR-424 and greater levels of miR-130a and miR-145. This study indicated the amount of urine exosomal miRNA changes in type 1 diabetes mellitus persons who have early DN, and miR-145 may be a promising biomarker for diabetes-related concerns [117].

#### 3.1.5. Diagnostic Function of Exosomes in Kidney Disorders

##### Acute Kidney Disease

Research using a rat model revealed that mice with ischemia/reperfusion-induced acute kidney injury (AKI) had reduced levels of urine exosomal AQP-1 and AQP-2 protein and mRNA. This result is important for identifying biomarkers and disease pathways [118,119]. In addition, various sets of exosomal microRNAs have the potential to be utilized as biomarkers for the classification of acute kidney damage progression related to ischemia/reperfusion injury. Conversely, the late stage of the damage (also referred to as the fibrotic stage) was when the elevations of miR-125 and miR-351 were seen. The most recent transcriptomics investigation revealed that mice with lipopolysaccharide-induced acute kidney damage had increased levels of exosomal miR-19b-3p, which is generated from renal tubular epithelial cells [120].

##### Chronic Kidney Disease

The roles of exosomes in chronic kidney disease (CKD), such as in types of renal fibrosis and DN, have been extensively studied. A recent research found that type 1 DN was associated with elevated urine excretion of prostasin, urokinase, and exosomal plasmin, along with the proteolytic activation of ENaC, which may have led to hypertension and dysfunctional Na+ excretion [121]. Additionally, in a proteomics study, urine exosomes from DN patients and healthy individuals were evaluated using a label-free quantitative method. Zubiri et al. [122] reported that the findings showed that DN patients had reduced voltage-dependent anion-selective channel protein 1 and increased urine exosomal bikunin precursor and histone-lysine N-methyltransferase. These results might enhance DN monitoring and diagnosis. Urine exosomes from DN patients were then found to contain higher amounts of AQP-2 and AQP-5, indicating that these proteins could be used as noninvasive biomarkers for DN diagnosis [123]. It was shown that urinary exosomal miR-320c, miR-6068, miR-1234-5p, miR-6133, miR-4270, miR-4739, miR-371b-5p, miR-638, miR-572, miR-1227-5p, miR-6126, miR-1915-5p, miR-4778-5p, and miR-2861 were increased in type 2 DN patients, while miR-30d-5p and miR-30e-5p were decreased [124].

##### Lupus Nephritis

A transcriptomics investigation on humans revealed that urine exosomes from individuals with active lupus nephritis (LN) had higher levels of miR-146a. According to this research, miR-146a may be utilized as an indicator of diagnosis to distinguish between systemic lupus erythematosus (SLE) patients without LN and those with active LN, as well as healthy controls [125]. According to a study by Solo et al. (2015), a reduction in urine exosomal miR-29c may also be utilized to foresee the initial stages of renal fibrosis and chronicity in patients with LN [126]. Tangtanatakul et al. (2019) research [127] found that urine exosomes taken from individuals with LN had downregulated levels of the anti-inflammatory microRNAs let-7a and miR-21. It may be possible to determine the clinical status of let-7a and miR-21 using these microRNAs as noninvasive biomarkers [128].

#### 3.1.6. Diagnostic Function of Exosomes in CNS Disorders

Exosomes are useful diagnostic tools because of all of their characteristics as well as their capacity to traverse the blood-brain barrier. Exosomes generated from the brain may really be thought of as a direct readout of the state of the central nervous system. Because the blood-brain barrier tightly regulates molecular transit, most of the floating proteins and nucleic acids come from sources other than neurons and are diluted in the bloodstream. Thus, the identification of circulating proteins, cytokines, or nucleic acids only provides an estimate of the body’s general level of inflammation. However, exosomes may be concentrated, which significantly raises the sensitivity of detection [129].

##### Exosomes in Neurodegenerative Disease

It was demonstrated that plasma exosomal α-synuclein is greater in Parkinson’s disease (PD) patients than in healthy controls [130]. Serum neuronal exosomal α-synuclein is substantially greater than in the amyotrophic lateral sclerosis (APS) group [130] and considerably less than in the multiple system atrophy group [131]. The gene expression of miR-223-3p and miR-7-1-5p targets and regulates the overexpression of α-synuclein [132]. Exosomes produced from plasma neurons in PD patients have significantly less vesicular glutamate transporter-1 (VGLUT-1), an intermediate in glutamate-synaptic interactions. However, excitatory amino acid transporter-2 (*EAAT-2*), another mediator in glutamate-synaptic connections, is upregulated [133,134]. PD patients have downregulated neurotrophic signaling pathway miR-19b-3p in CSF exosomes, while upregulated miR-153, miR-409-3p, miR-10a-5p, and let-7g-3p [135].

It is possible to evaluate the bulk of proteins present in exosomes using proteomic analysis [136]. As the authors put it, exosomes are a kind of “brain fluid biopsy” since they can be separated from the CSF. Prionogenic proteins, such as amyloid precursor protein APP, are among these proteins. There are presently few studies showing the diagnostic use of exosomes. Goetzl et al., however, recently showed that blood-derived exosomes include proteins such as heat shock protein 70, lysosome-associated membrane protein-1 (LAMP-1), and cathepsin D, whose levels are changed in preclinical AD years before the disease onset [137]. Additionally, exosomes from the CSF of AD patients were extracted by Saman and colleagues, who demonstrated that these exosomes had higher amounts of AT270 phospho-tau than exosomes from healthy controls [138]. After separating neuron-derived exosomes from human plasma, Fiandaca et al. demonstrated that exosomal levels of two particular phosphor-tau, P-T181-tau and P-S396-tau, increased in a collection of individuals who were cognitively normal at first but developed AD ten years later [129]. Their results showed no difference in exosomal phospho-tau levels between the pre-clinical AD and AD groups. Additionally, they proposed that because exosomal levels of this protein are higher in pre-clinical AD patients and much higher in symptomatic AD, Aβ1-42 may be a valuable biomarker for the progression of the disease.

Minagar et al. demonstrated that when multiple sclerosis (MS) patients had remission, their plasma levels of exosomes generated from endothelial cells decreased to control levels [139]. Additionally, Giovanelli et al. showed that exosomes extracted from specimens of blood and urine in MS patients using natalizumab medication included miR-J1-3p and miR-J1-5p, suggesting that drug treatment outcomes in MS patients may be monitored [140]. 

Various miRNAs related to MS have been found in serum-derived exosomes, and their profiles seem to indicate various disease subtypes. Relapsing-remitting multiple sclerosis (RRMS) patients have been found to have miRNA-15b-5p, miRNA-451a, miRNA-30b-5p, and miRNA-342-3p; secondary progressive multiple sclerosis (SPMS) patients have been found to have miRNA-127-3p, miRNA-370-3p, miRNA-409-3p, and miRNA-432-5p [141]. Many studies have shown the role of miRNAs in CNS immunomodulation, which makes sense given the T cell-mediated autoimmune nature of MS. It has been demonstrated that MS patients have higher levels of exosomal miRNA let-7i [142].

##### Stroke

Within the setting of cerebrovascular diseases, exosome content may serve as a useful biomarker. For example, it has been shown that people who have had an acute ischemic stroke have higher serum concentrations of exosomes and exosomal concentrations of brain-specific microRNAs (miR-9 and miR-124) than the control group. Additionally, it has been shown that the levels of miR-9 and miR-124 positively correlate with the infarct sizes, the serum levels of IL-6, and the rating on the National Institutes of Health Stroke Scale (NIHSS) [143]. This implies that exosomes may potentially be utilized to assess the extent of ischemic injury-related damage. Exosomal miR-223 amounts were shown to be greater in individuals who had experienced an acute ischemic stroke than in the control group by Chen and colleagues [144]. Additionally, there was a significant association between this microRNA’s level and NIHSS scores, and it grew steadily until 72 h following the stroke [144]. It should be noted that the incidence of acute ischemic stroke, the severity of the stroke, and the short-term results were all associated with a higher quantity of exosomal miR-223 [144].

##### Neuropsychiatric

Schizophrenia (SCZ) is the most prevalent mental illness. The symptoms of SCZ include delusions, hallucinations, and cognitive and emotional abnormalities. The condition often manifests between the ages of 12 and 20. In SCZ, exosome indicators in several pathways have been found. SCZ patients’ prefrontal cortex exosomes had higher levels of miR-497 [145]. miR-497 is a member of the miR-15/107 family of microRNAs, which have been linked to neurodegenerative disorders and have an impact on cortical gene expression [146]. Patients with SCZ have changed redox-related molecules [147,148]. Antioxidant protein DJ-1 controls the expression of genes involved in antioxidant defence, shielding cells from the damaging effects of oxidative stress [149,150]. Serum exosomes from patients with SCZ had substantially more DJ-1 and significantly less miR-203a-3, which targets the mRNA of DJ-1 [151]. Additionally, neuronal plasma exosomes from patients with drug-naïve first-episode SCZ had lower ratios of phosphorylated AKT1/2/3, GSK-3β, mTOR, and p70S6K to total proteins, as well as phosphorylated mTOR (pS2448-mTOR) to total m-TOR [152].

Anxiety, mania, bipolar affective disorder (BD), and depression are examples of mood disorders [153]. Every exosomal biomarker that has been found has a role in neurodevelopment in depressed individuals. Serum exosomes from patients with major depressive disorder (MDD) have considerably higher levels of miR-186-5p, miR-3122, and miR-4428, which target serpin family F member 1 (*SERPINF1*) while having substantially decreased amounts of SERPINF1/pigment epithelium-derived factor, a form of neuronal trophic factor [154]. miR-146a-5p is significantly upregulated in MDD serum exosomes [155]. The pathophysiology of depression has been associated with reduced glutamate levels in certain brain areas [156,157]; the plasma exosomes of people with treatment-resistant depression have significantly higher levels of miR-335-5p [156]. To control neuronal excitability, miR-335-5p targets glutamate metabotropic receptor 4 (*GRM4*) [158]. Additionally, intriguing research reveals that there is a large increase in miR-29c in the exosomes of BD patients in the prefrontal cortex [159]. Deniz Ceylan et al. found in another research that whereas miR-185-5p is highly elevated in plasma exosomes from BD patients, miR-484, miR652-3p, and miR-142-3p are dramatically reduced [160].

#### 3.1.7. Diagnostic Function of Exosomes in Pregnancy Disorders

Exosomes have a lot of potential for use in prenatal screening and the identification of pregnancy problems, including hypertension and hyperglycemia during gravidity. Although their utilization in maternal peripheral blood, urine, and amniotic fluid during pregnancy is still being studied, exosomes are a noninvasive and promising technique for the early detection and prognosis of pregnancy problems. This is due to the knowledge that placental cells may interact with the mother’s body by releasing exosomes and that the microenvironment controls the production and release of placenta-derived exosomes (PDEs), including oxygen tension and glucose content [161].

##### Exosomes in Hypertensive Disorder of Pregnancy

Preeclampsia (PE) may induce placental hypoxia, which causes placental cells to secrete more exosomes and change their composition [162]. For the extraction and measurement of PDEs, placental alkaline phosphatase (PLAP) is a placenta-specific marker in addition to standard markers [162,163]. According to Pillay et al., individuals with early-onset and late-onset PE had a considerably lower ratio of PDEs to total exosomes (exosome containing PLAP ratio), even if their relative concentration of PDEs was much greater than that of those with normotension [164]. Biro et al. collected plasma samples from healthy controls and pregnant women with PE, gestational hypertension, or chronic hypertension [165]. Next-generation sequencing was utilized to find exosomal miRNAs in Carlos Salomon’s work on placental exosome alterations in PE women throughout gestation. The two miRNAs selected may be potential predictors of PE incidence, which might greatly enhance the treatment of pregnant hypertension. The research also found that exosome concentrations of miR-486-1-5p and miR-486-2-5p were considerably greater in PE than in normal controls [166].

##### Exosomes in Prenatal Screening

Identification of exosomes may also be helpful in prenatal screening. The exosome contains PLAP proportion from maternal plasma, which was considered a potential biomarker of fetal development and placental function since it was significantly reduced in patients with fetal growth restriction than in healthy controls [167]. Identification of plasma exosomes may allow for far more precise diagnosis and prenatal monitoring of fetal growth restriction when combined with type-B ultrasonography and physical examination. Moreover, amniotic fluid-derived exosomes with diminished miR-300 and miR-299-5p might be used as biomarkers for the diagnosis of congenital obstructive nephropathy. Unfortunately, invasive amniocentesis was still practiced [167]. In conclusion, the most widely researched biomarkers for pregnancy problems, prenatal screening, and preterm birth monitoring are maternal circulating exosomal miRNAs, especially PDEs [168]. Maternal alterations in gestation also had an impact on urinary exosomes, which might be used to diagnose hypertension and intrahepatic cholestasis [169]. In addition to standard blood and urine testing during pregnancy, exosomal biomarkers can be utilized to detect and predict issues in both pregnant mothers and their unborn children [170].

#### 3.1.8. Diagnostic Function of Exosomes in Organ Transplantation

It has been shown that the transfer of EVs from the donor graft organ, which initiates an immune reaction to allografts and transfers donor MHC to recipient antigen-presenting cells, is necessary for allograft rejection [171]. The T cell activation method via exosomes is referred to as the semidirect approach [172,173]. Clinical transplantation prevention and therapy depend on early, accurate diagnosis and long-term immunologic rejection monitoring. Immune rejection may be detected by current detection techniques, but by that time, the transplanted organ has often suffered irreparable harm. Because chronic allograft rejection affects the lives of most transplant patients, there is an urgent need for simple, noninvasive techniques to detect it. Therefore, the presence of donor-specific exosomes and exosomal changes caused by immunologic rejection over time may serve as a surrogate biomarker for acute or chronic rejection of solid organ allografts [174].

In the context of lung transplantation, serum/BALF-based exosomes containing recipient immunoregulatory miRNAs, donor HLA, and SAgs may be involved in acute rejection and may identify allograft rejection early in the transplantation process [175]. Furthermore, patients who received lung transplants either with or without acute rejection showed a substantial difference in the analysis of BALF-based exosomal mRNAs, and the increased molecules in acute rejection samples showed a strong propensity toward an inflammatory milieu associated with innate as well as adaptive immune responses [176]. According to Kennel et al. [177], patients of heart transplants who had allograft rejection had varying levels of circulating exosomal protein content. They also found that 15 proteins were notably distinct and mostly associated with the immune response. As a result, exosomal protein analysis may prove to be an effective post-transplant monitoring technique. Furthermore, by introducing exosomes that included miR-142-3p from heart transplant recipients to endothelial cells, a study presented a novel understanding of the mechanism of acute cellular rejection in cardiac allografts by transferring exosomes containing miR-142-3p from heart transplant recipients to endothelial cells, which compromised endothelial barrier function by downregulating RAB11 family interacting protein 2 (RAB11FIP2) [178]. In total, 169 urine exosome proteins were found using nano-ultra performance liquid chromatography-tandem mass spectrometry (nano-UPLC-MS/MS); of these, 17 were overexpressed in acute rejection patients, and 46 were upregulated in stable receivers. Lastly, they chose hemopexin and tetraspanin-1, which were markedly increased in acute rejection patients, as possible indicators for diagnosing acute rejection in kidney transplantation users. Kidney transplantation also involves plasma exosomes [179]. According to Zhang et al. [180], measurement of plasma exosome mRNA may be a useful method for kidney transplantation patients’ early allograft rejection diagnosis. Furthermore, it was shown that patients who had acute rejection after liver transplantation had higher levels of blood of hepatocyte-derived exosome-associated miR-122, miR-148a, and miR-194. Additionally, it was noted that these levels appeared to increase before aminotransferase levels did [181].

### 3.2. Diagnostic and Prognostic Function of Exosomes in Malignant Disorders

#### 3.2.1. Diagnostic Function of Exosomes in Brain Tumor

The investigation of exosomal hsa-miR199a-3p [182] is one of the most recent discoveries made in relation to exosomes in neuroblastoma (NB). The overexpression of exosomal hsa-miR199a-3p has been shown to be related to a high-risk scenario and a poor prognosis for NB. Recent research has shown that a high level of exosomal miR-375 is associated with bone marrow (BM) metastases in patients with NB [183]. Therefore, exosomal miR-375 may be an essential diagnostic biomarker in identifying the course of BM metastatic disease, and it may also offer a novel potential target for patients with NB who have BM metastases [183]. Exosome-derived miR-21 was one of the first miRNAs that was suggested for use in diagnostic situations involving patients with glioblastoma multiforme (GBM). Glioma cell invasiveness, migration, and tumor grading are all modulated by miR-21, which is often overexpressed in malignant gliomas [184,185]. The quantity of microRNA-21 found in exosomes in the CSF of GBM patients was 10 times higher than that found in the exosomes of subjects who were healthy. Similarly, serum exosomes from GBM patients had 40 times more miR-21 than those from healthy people. The contents of glioma-derived exosomes (GDE), which include miR-21, miR-222, and miR-124-3p, can now be identified using minimally invasive procedures to facilitate the identification of brain tumors, the estimation of glioma pathological grading, and the discovery of preoperative metastases [184,185].

#### 3.2.2. Diagnostic Function of Exosomes in Thoracic Tumor

##### Lung Cancer

Kristine et al. [11] developed an EV array that paired 37 antibodies that target proteins associated with lung cancer with a panel of CD9, CD63, and CD81 antibodies to examine circulating exosomes from both healthy people and lung cancer patients. The authors’ integrated 30-marker model EV array was able to distinguish between the two groups with an accuracy rate of 75.3%. It was demonstrated that the exosomal protein of New York esophageal squamous cell carcinoma-1 (NY-ESO-1) maintained a significant concentration-dependent impact on worse survival after further testing using the Bonferroni correction procedure [186]. Their approach detected and carefully contrasted the proteome patterns of saliva and serum exosomes from lung cancer patients as well as healthy controls using LC-MS/MS [187]. It is possible to differentiate between those with lung inflammation and those with lung adenocarcinomas by looking at the amounts of exosomal RNA expression in pleural effusion. Compared to benign lung inflammation, lung cancer exhibited considerably greater gene expression for a panel including miR-200b, miR-200c, miR-141, and miR-375. These miRs also have values greater than 0.95, according to ROC curve analysis. However, the most promising exosomal transcript for diagnosis was lipocalin-2 (LCN2), which had an AUC of 0.99 [188] Pleural effusion collection is a bit more invasive than blood collection, despite the fact that this is a potentially useful diagnostic. It would be interesting to see whether these results could be repeated using blood from people with lung inflammation rather than lung cancer. For instance, several research has shown that exosomal miRNAs are the most promising biomarkers for detecting early lung cancer in body fluids because of their reliability, accessibility, and specificity. As an illustration, Sun, Z. et al. found that miR-96 significantly improved lung cancer diagnosis [189].

##### Breast Cancer

Proteins on the outside and inside of exosomes may also be used as cancer indicators. It is interesting to note that Wang and colleagues [190] recently showed that exosomal tetraspanin CD82 was much more abundant in the serum of breast cancer (BC) patients than in healthy controls and that CD82 expression rose sharply as malignant breast cancer progressed. Furthermore, urine exosomal tetraspanin CD63 and miR-21 expression together showed a 95% sensitivity to early BC identification, despite the fact that neither marker is specific to BC [191]. Researchers found over 100 phosphoproteins in plasma exosomes using LC-MS/MS, which are noticeably more prevalent in breast cancer patients than in healthy controls. Furthermore, they used a quantitative mass spectrometry (MS) method called parallel reaction monitoring (PRM) to confirm that four phosphoproteins—tight junction protein 2 (TJP2), nuclear transcription factor (NFX1), Ral GTPase-activating protein subunit alpha-2 (RALGAPA2), and cGMP-dependent protein kinase 1 (PKG1)—were significantly upregulated in patients with breast cancer [42]. Furthermore, the plasma exosome’s concentration of human epidermal growth factor receptor-2 (HER2) was almost identical to that seen in tumor tissues [192]. In a noninvasive manner, the exosomal HER2 in circulation may represent the molecular categorization of the tumor tissues. Exosomal fibronectin and *EGF like repeats and discoidin domains 3* (*EDIL3*) levels were markedly elevated in breast cancer patients compared to controls and sharply decreased following tumor resection, indicating that they could be crucial prognostic and diagnostic indicators for patients with breast cancer [193,194]. In a recent study, Kibria et al. [195] profiled the protein expression of exosomes extracted from BC patients’ cell lines and blood as well as healthy controls using an automated microflow cytometer. When comparing BC patients’ circulating exosomes to controls, they discovered a substantial drop in CD47 expression. Interestingly, CD47 is a surface protein linked to cancer whose expression stops the innate immune system from identifying cancer cells, which promotes the growth of tumors [196]. Other intriguing research showed that BC patients had greater levels of serum exosomal-annexin A2 (exo-AnxA2) than females without cancer, particularly for triple-negative BC (TNBC) as opposed to luminal and HER2-positive BC. Furthermore, poor overall survival, poor disease-free survival, and tumor grade were all substantially correlated with high levels of exo-AnxA2 expression in BC. Additionally, this research demonstrated that exo-AnxA2 stimulates angiogenesis. Consequently, exo-AnxA2 is a possible therapeutic target and prognostic biomarker for TNBC [197]. Circulating exosomal miRNA was recently examined as a possible breast cancer biomarker [198].

#### 3.2.3. Diagnostic Function of Exosomes in Gastrointestinal Tumor

##### Pancreatic Cancer

The majority of pancreatic cancer types are composed of pancreatic ductal adenocarcinoma (PDAC). The five-year survival rate for this extremely aggressive and metastatic tumor type is 7% [199]. Invasive surgery and imaging tests are now used to confirm the diagnosis of pancreatic cancer. As of right now, sialyl Lewis (a) blood group 19-9 (CA19-9) is the only serum marker test available for pancreatic cancer. However, the sensitivity and specificity of CA19-9 vary, ranging from 60% to 90% and 68% to 91%, respectively [200]. Finding novel and early biomarkers for the diagnosis of pancreatic cancer is thus essential for a favorable patient outcome. The cell-surface proteoglycan glypican-1 (GPC1) is more abundant in exosomes produced from pancreatic cancer cells. It was first shown to be a potential target for the early detection of pancreatic cancer [9]. Exosomal GPC1 cannot differentiate pancreatic tumors from non-tumorous controls, according to Lai and colleagues [113], and levels of exosomal GPC1 were only marginally reduced after resection. Zhou et al. carried out a follow-up investigation to ascertain the effectiveness of GPC1 as an early marker. After five years of monitoring PDAC patients, they discovered that CA19-9 was noticeably more effective than GPC1 in differentiating PDAC patients from healthy individuals. They did note, however, that a worse overall survival rate was associated with elevated blood levels of GPC1. They concluded that GPC1 might be a prognostic indicator rather than a valid diagnosis. Another study used a panel of exosomal mRNA and short nucleolar RNA (snoRNA) to investigate CA19-9’s ability to distinguish between patients with pancreatic cancer and healthy controls. This study found that the amounts of *WAS protein family member 2 (WASF2)*, *ADP ribosylation factor 6 (ARF6)*, *small nucleolar RNA, H/ACA*
*Box 74A (SNORA74A)*, *small nucleolar RNA*, and *H/ACA Box 25 (SNORA25)* could distinguish between patients with pancreatic cancer and healthy patients, with a ROC curve value greater than 0.90 in comparison to CA19-9. Out of their four-gene panel, they found that *WASF2* had a greater connection with the risk of pancreatic cancer, even when compared to CA19-9. Therefore, exosomal *WASF2* might offer a new way to assess a patient’s risk of developing pancreatic cancer [201].

##### Colorectal Cancer

Colonoscopy is often used for regular screening in order to diagnose colon cancer. This approach involves a patient abstaining from solid food intake and consuming a preparation that thoroughly evacuates the contents of the colon. The patient may find this procedure unpleasant and painful. One advantage of a colonoscopy is that if the doctor finds polyps in the colon, they may be removed, which will stop a tumor from growing. Sadly, only around 40% of colorectal cancers (CRC) are discovered at this early stage. Therefore, more techniques are required to guarantee a thorough screening for colorectal cancer [202]. Usually seen throughout fetal gastrointestinal development, CEA resurfaces in the blood during gastrointestinal tract cancer. Currently, hospitals utilize it as a marker to keep track of patients with colon cancer [203]. There is, however, space for new, more precise markers since CEA is not exclusive to colon cancer. Exosomes were examined by Lee et al. for the presence of distinctive protein markers [203]. To check for the existence of distinctive protein markers, they used the ATCC colon epithelial cell line CRL-1541, as well as the colon cancer cell lines HT-29 and HCT-116. Exosomal tetraspanin 1 (TSPAN1) was shown to be a promising marker for the identification of colon cancer. This protein showed a 75.7% sensitivity in patient plasma samples. TSPAN1 and CEA together could provide a novel diagnostic marker for the diagnosis and prognosis of colon cancer. There have also been reports of GPC1 being used as a CRC diagnostic marker. According to the research, exosomes from tumor tissues and plasma of CRC patients had substantially lower percentages of GPC1+ exosomes and GPC1 protein expression after surgery than exosomes from peritumoral tissues and plasma of healthy persons [204].

##### Hepatocellular Carcinoma

Sohn et al. found that individuals with chronic hepatitis B infection had considerably higher levels of expression of miR-18a, miR-221, miR-222, and miR-224 in their exosomes than those suffering from HBV-associated hepatocellular carcinoma (HCC). This implies that the four miRNAs listed above may be novel plasma indicators for liver cancer detection [205]. Additionally, there was a strong correlation between exosomal miR-30d, miR-140, and miR-29b with the overall survival of HCC patients. Accordingly, exosomal miRNAs could potentially serve as predictive biomarkers for HCC and help decide the best course of treatment for advanced hepatocellular carcinoma [206]. Li et al. claimed that long non-coding RNA (lncRNA) may also be found in plasma in addition to miRNA and that the protective function of exosomes may be one of the reasons for its consistent presence in blood [207]. High blood exosome circRNA levels may be able to differentiate HCC patients from healthy people, according to a recent study by Li et al. [207]. Exosomes may develop into a potential marker for the diagnosis of HCC, according to the findings above. Exosomal miR-665 levels in patient serum were directly correlated with tumor size; also, a higher level of exosomal miR-665 indicated a shorter survival time and a worse prognosis in the HCC group. RNA LINC00161 showed a similar pattern in urine-derived exosomes from people with HCC compared to healthy controls. The study yields an AUC of 0.794 with sensitization and particular characteristics of 75.0% and 73.2%, respectively [208].

#### 3.2.4. Diagnostic Function of Exosomes in Thyroid Cancer

Notably, individuals with thyroid cancer (TC) who had a high risk of recurrence did not have elevated blood thyroglobulin levels after surgery. The U-Ex-Tg levels, however, were noticeably higher. Accordingly, U-Ex-Tg may serve as a more accurate molecular indicator for prognosis and the prediction of postoperative recurrence in differentiated TC (DTC) [209]. Additionally, it was shown that serum-purified EVs (SPEs) from individuals with papillary TC (PTC) had greater levels of HSP27, HSP60, and HSP90 than those in BG and peritumoral tissues (PT). Compared to PTC after surgery, these SPEs indicators from preoperative PTC were greater. Furthermore, it is probable that EVs encased these markers, which subsequently took part in the intercellular communication of TC, since they were found in typical locations, such as the cytoplasm in benign goiter (BG) samples, but closer to the plasma membrane in PTC tissues [210]. According to earlier research, the overexpressed miR-146b, miR-222 [211], miR-5189-3p [212], miR-31-5p, miR-21-5p [213], miR-346, miR-10a-5p, miR-34a-5p [214], hsacirc_007293, hsacirc_031752, hsacirc_020135 [215], and miR-4433a-5p [216] in EVs function as diagnostic biomarkers for PTC and follicular thyroid carcinoma (FTC), allowing for the identification of TC and its differentiation from benign or healthy thyroid nodules [217]. The benefits of employing numerous biomarkers for diagnosis have been reported in earlier research. PTC and FTC may be distinguished with 100% sensitivity and 87% specificity when miR-21-5p and miR-181a expression in SPEs are analyzed together [213]. With 74% AUCs, 100% sensitivity, and 87% specificity [218], the combined study of miR-16-2-3p, miR-223-5p, miR-101-3p, and miR-34c-5p expression in SPEs may distinguish PTC from BG, producing higher values than those predicted using individual biomarkers.

#### 3.2.5. Diagnostic Function of Exosomes in Urogenital Tumor

##### Prostate Cancer

Exosomes from human serum were shown to include CD9 and the cell surface enzyme gamma glutamyl transferase 1 (GGT1). GGT1 expression and serum exosomal GGT activity were considerably greater in prostate cancer (PCa) patients than in benign prostatic hyperplasia (BPH) individuals [219]. This might be a novel diagnostic indicator to distinguish between these two conditions. The greatest diagnostic value was found by comparing the proteome of the urine exosomes of PCa patients with that of healthy participants [220]. As independent biomarkers, a number of urine exosome proteins showed great sensitivity and specificity for prostate cancer; when combined in a multi-panel test, they may fully differentiate prostate cancer from non-disease controls [220]. According to Liu et al., exosomes from prostate cancer have a substantial prostate-specific antigen (PSA) enrichment, which is indicative of the original PCa cells [221]. MiR-1290 and miR-375 were identified as possible predictive markers in castration-resistant prostate cancer (CRPC) from plasma vesicles collected from prostate cancer utilizing the precipitation-based ExoQuick technique because their levels are linked to a poorer overall survival (*p* < 0.004) [222,223]. Those with CRPC with the exosomal androgen receptor splice variant (AR-V7) exhibited lower levels of sex hormones and a poorer prognosis, according to Joncas et al. [224]. Recently, it was shown that miR-1246, an exosomal miRNA obtained from blood, may be a potential biomarker for PCa. With an AUC of 0.926, 100% specificity, and 75% sensitivity [225], the researchers discovered that miR-1246 could be a reliable indicator of aggressive prostate cancer. The levels of exosomes could play an important role in providing the diagnostic and prognostic overview of prostate cancer. Exo-PSA consensus score (EXOMIX), immunocapture-based ELISA (IC-ELISA), and nanoscale flow-cytometry (NSFC) demonstrated 98% to 100% specificity and sensitivity for BPH-PCa discrimination, while statistical analysis revealed that the levels of plasmatic exosomes expressing both CD81 and PSA were significantly higher in PCa compared to both BPH and healthy donors, reaching 100% specificity and sensitivity in differentiating PCa patients from healthy individuals [226]. Similarly, exosomal CA IX expression levels and activity may be used as a biomarker of PCa cancer development [227]. Another study showed that compared to urological disease, PCa had substantially greater plasmatic levels of exosomes, and the exosomes were smaller. The study introduced a noninvasive exosome-based clinical approach for prostate cancer follow-up and early diagnosis, potentially serving as a screening test [228].

##### Bladder Cancer

As bladder cancer is a diverse illness marked by a high mutation load, multiple studies emphasized the need for integrated molecular patterns better than single gene testing, which may be faulty in certain tumors but not others. Huang et al. used RNA sequencing to find an RNA panel made up of two lncRNAs (MIR205HG and GAS5) and three mRNAs (*KLHDC7B*, *CASP14*, and *PRSS1*) that can differentiate bladder cancer patients from healthy participants (AUC = 0.924, 95% CI, 0.875–0.974). The amounts of expression of these five RNAs were also linked to clinicopathological characteristics [229]. The amounts of expression of an exosome lncRNA panel (UCA1-201, UCA1-203, MALAT1, and LINC00355) were also shown to have excellent sensitivity and specificity in distinguishing urothelial cancer from healthy tissue (92% sensitivity and 91.7% specificity) in research by Yazarlou et al. (*n* = 108) [230]. Even in individuals with negative cytology, urine exosomal miR-21-5p was able to distinguish between those with urothelial carcinoma (AUC = 0.9, sensitivity, 75.0%; specificity, 95.8%), according to another research (*n* = 60) conducted by Matsuzaki et al. [231]. According to Piao et al., patients with bladder cancer had a significantly higher expression ratio of miR-6124 to miR-4511 than patients with hematuria or pyuria (sensitivity: 91.5%; specificity: 76.2%). In patients with gross hematuria, the sensitivity even rose to 94.0% [232]. These results are important because they have found biomarkers that, with higher sensitivity and specificity than cytology, can determine which hematuria patients need a complete workup.

##### Kidney Cancer

Exosomes have emerged as a novel source of noninvasive tumor biomarkers, in addition to hypoxia-inducible factor 1 subunit alpha (HIF-1a) [233]. It was shown that renal cell carcinoma (RCC) patients’ urine EVs included lower levels of mRNA for *pterin-4 alpha-carbinolamine dehydratase-1 (PCBD1)*, *glutathione transferase alpha 1 (GSTA1)*, and *CCAAT enhancer binding protein alpha (CEBPA)* than controls. One month after nephrectomy, however, the mRNA levels of these three genes reverted to normal [234]. Zhang et al. found that serum samples from patients with clear cell RCC had higher levels of exosomal miR-210 and miR-1233 than healthy controls and that these levels decreased following nephrectomy. The authors came to the conclusion that exosomal miR-210 and miR-1233 could be useful markers for low-invasive diagnosis and follow-up with patients with clear cell RCC [235]. Notably, miRNAs such as miR-29a, miR-650, and miR-151 were connected to tumor invasion and metastasis. Another research team analyzed exosomal miRNAs isolated from plasma samples and discovered that miR-92a-1-5p was considerably downregulated and that miR-149-3p and miR-424-3p were increased [236]. Exosomes are rich in Long non-coding RNA (lncRNAs), which are crucial for the growth, division, invasion, and dissemination of cancer cells [237]. Research has shown that exosome-mediated transfer of lncARSR enhances the expression of *AXL* and *c-MET* in RCC cells by competitively binding to miR-34/miR-449, hence aiding the cells’ acquisition of sunitinib resistance [238].

##### Ovarian Cancer

Exosomes extracted from various bodily fluids and tissue samples are examined as potential sources of biomarkers for the diagnosis and prognosis of ovarian cancer. Serum-derived exosomes from patients with ovarian cancer had higher levels of miR-200c, miR-145, and miR-93 than those from patients with benign illness and borderline ovarian cancer, according to an analysis of a collection of miRNAs that are overexpressed in ovarian cancer [239]. Compared to stage I–II patients, stage III–IV individuals (which involves those with lymph node metastases) have significantly greater expression of miR-200b and miR-200c [240]. Exosomes from patients with epithelial ovarian cancer exhibit greater levels of miR-146b-5p expression than those from the healthy control group [241]. Because they are readily available samples, exosomes in urine have recently caught the interest of researchers [242]. Patients with ovarian cancer reported greater levels of miR-92a expression in urine-derived exosomes [243].

##### Cervical Cancer

The level of survivin expression in cervical cancer (CC) cells is rather high. It has been shown that the concentration of survivin is very high inside the exosomes that are produced from cancer cells [244]. Exosomal miR-221-3p is another biomarker for cancer that promotes metastatic potential. This biomarker works by downregulating *mitogen-activated protein kinase 10 (MAPK10)*, which contributes to the development of cancer. There is also the possibility that exosomes that include the genes for *activating transcription factor 1 (ATF1)* and *RAS* might be used as biomarkers for cancer. The expression of these exosomal cargoes was increased in a humanized tumor mouse model of cancer, and as a result, they have the potential to serve as biomarkers for cancer [245].

#### 3.2.6. Melanoma

One putative biomarker for melanoma is a cell surface proteoglycan termed chondroitin sulfate proteoglycan 4 (CSPG4) [246,247]. The expression of CSPG4 in tumor-derived exosomes (TEXs) was shown to be 19 times higher in patients with melanoma as compared to the healthy control group in research that was conducted by Monika and colleagues [248]. Annexin A1 and annexin A2 could be other biomarkers that are necessary for the invasion of melanoma by signaling proliferation. TEXs release from malignant melanoma cells have a high level of expression with annexin A1; however, the expression of annexin A2 was shown to be downregulated [249]. It has been shown that melanoma patients have greater levels of expression of serum exosomal microRNAs, including Exo-miRNA-532-5p and Exo-miRNA-106b, compared to healthy individuals. It may be possible to distinguish between patients with early-stage and late-stage melanoma by using these exo-miRNAs as melanoma biomarkers. Additionally, samples were obtained from 25 melanoma patients and 25 healthy individuals for the study. This miRNA panel detected 22 out of 25 normal individuals (sensitivity of 88.0%) and 23 out of 25 melanoma cases (sensitivity of 92.0%) [250]. Additionally, Logozzi et al. provided a novel noninvasive method that enables the identification and measurement of human exosome levels in melanoma patients’ plasma. According to their findings, the in-house sandwich ELISA (Exotest) for identifying plasma exosomes with tumor-associated antigens could be an innovative clinical management tool for cancer patients [251]. Given the existence of these markers, exosomes may prove to be a very useful tool for melanoma diagnosis, prognosis, and therapy.

#### 3.2.7. Hematologic Malignancy

##### Leukemia

In patients with acute myeloid leukemia (AML), serum exosomal miR-10b is a factor that operates independently as a predictive component for overall survival. There is a significant rise in the levels of miR-10b in patients with AML [250]. Patients with AML have greater serum levels of miR-10b, and a worse prognosis is strongly associated with higher levels of this gene. AML may thus be diagnosed and prognosed using serum exosomal miR-10b. M1 had high expression levels of miR-146a/b, miR-181a/b/d, miR-130a, miR-663, and miR-135b, while M5 had high expression levels of miR-21, miR-193a, and miR-370 [252]. Furthermore, it has been shown that peripheral blood mononuclear cells from patients with adriamycin-resistant AML cell lines and multi-drug-resistant AML have downregulated miR-155 and that this downregulation has a positive correlation in these patients. MiR-155 may be used as a very sensitive monitoring indicator to track drug resistance and micro-residual focus [252]. Exosomes derived from AML are abundant in CD33, CD34, and CD117, and their overall protein content is significantly higher than that of healthy controls. The content of certain proteins, such as TGF-b1, decreases during the initial diagnosis and effective treatment of AML. As a result, these exosomes can be utilized to detect leukemia relapse and the status of drug resistance [253].

##### Lymphoma

In the context of lymphoma, a potential biomarker could be the downregulation of exosomal mir-451a, according to the findings of several studies [254,255]. Taking into consideration the fact that each of these studies evaluated more than 59 diffuse large B-cell lymphoma (DLBCL) samples, it is important to remark that their findings are more trustworthy than those of previous investigations [255,256]. A number of intriguing findings demonstrated that miR-181a-5p, miR-181b-5p, and miR-181d-5p were down-expressed; hence, more evaluations could be conducted to determine if any of the isoforms of exosomal mir-1815p are downregulated in DLBCL [255]. Furthermore, the search results showed that there is no discernible difference between DLBCL patients and controls in terms of exosomal miR-146a expression [257]. In view of the fact that circulating miRNAs are capable of being embedded in exosomes, there is a need for more research to validate the diagnostic potential of exosomal miR-146a in DLBCL [258]. Additionally, lower levels of exosomal miRNA-107 and miR-451a indicated a bad prognosis in DLBCL, whereas greater amounts of exosomal miR-125b-5p and miR-99a-5p were associated with a shorter progression-free survival (PFS) [256,259]. A novel exosomal miRNA profile was discovered by Yeh et al. in plasma samples taken from patients with CLL. This profile includes miR-29 family, miR-150, miR-155, and miR-223 [260].

Appendix A represents the capacity of exosomal diagnostic and prognostic biomarkers for malignancy disorders.

#### 3.2.8. The Advantage of Exosomes for Cancer Early Detection

Exosome research has the potential to lead to the discovery of new biomarkers and therapeutic targets, which would further our knowledge of the biology of cancer [261]. It is important to note that exosomes provide a number of benefits for the early identification of cancer, which makes them a prospective route for enhancing the accuracy and efficacy of diagnostic procedures. Exosomes, for instance, may be collected at several time periods, which enables dynamic monitoring of the course of the illness and the response to therapy [262]. Changes in the molecular cargo of exosomes over time may guide therapy changes, offering a real-time perspective of the illness [263,264]. The following are some of the primary benefits of using exosomes for the early diagnosis of cancer.

##### Specificity and High Sensitivity

Exosome-based assays represent a significant advancement in the area of cancer diagnostics because of their exceptional sensitivity and specificity [265]. The likelihood of getting false-positive findings is greatly decreased since these tests are so good at detecting compounds unique to cancer. This precision guarantees a more patient-centric approach to healthcare by lowering the amount of unnecessary diagnostic procedures as well as the anxiety that false alarms often cause [265]. Logzzi et al. found that specific exosomes of the PSA could distinguish among patients with PCa, patients with BPH, and healthy controls more successfully than the traditional blood PSA test. When it came to distinguishing between PC and BPH, immunocapture-based ELISA (IC-ELISA) attained a sensitivity of 98.57% and a specificity of 80.28% [265]. According to Saad et al. [266], the ability of exosome-based assays to identify even minute quantities of cancer biomarkers, even when these biomarkers are present in low concentrations, is what sets them apart from other types of assays [267]. This increased sensitivity becomes a powerful instrument in the early diagnosis of cancer, which has the ability to diagnose the illness before clinical signs reveal themselves. The early identification that is made possible by assays that are based on exosomes significantly improves the likelihood of effective treatment results, which eventually leads to an improvement in the patient’s prognosis.

Sophisticated analytical techniques are needed due to the vast volume of data and the intricacy of the information obtained from exosome profiling. Bioinformatics and machine learning are crucial to this approach. Machine learning techniques are especially useful for the analysis of exosome profiles due to their exceptional ability to identify trends in large datasets [268]. A major advancement in early cancer detection techniques is made by Li et al., machine learning approach for noninvasive cancer diagnosis using exosome protein markers, which achieves excellent precision in identifying cancer types with a modern biomarker signature and sophisticated data models [269]. Wang et al. used the Least Absolute Shrinkage and Selection Operator (LASSO) regression technique to develop a prediction model based on differentially expressed genes (DEGs). The outlook assessment and receiver operation characteristic curve analysis were used to validate the model’s anticipated accuracy and sensitivity [270]. The algorithms in question are able to distinguish between malignant and non-cancerous samples, categorize various forms of cancer, and forecast the results of treatment processes. Additionally, bioinformatics tools are necessary for the management and interpretation of the vast amounts of data that are generated by exosome analysis. These tools assist in the process of gaining useful insights from the intricate molecular profiles of exosomes. A paradigm change has occurred in the field of cancer diagnosis as a result of the merging of liquid biopsies, nanotechnology, machine learning, and bioinformatics. Through the use of exosome analysis, this convergence makes it possible to perform noninvasive, very accurate early cancer diagnosis, as well as dynamic disease monitoring [270].

##### Noninvasive Sample and Patient Friendly

As a result of their ability to be easily separated from body fluids that are easily accessible, such as blood, urine, saliva, and cerebrospinal fluid [271], exosomes promise a groundbreaking avenue for noninvasive diagnostic practices [272]. This noninvasive sampling technology reduces the need for unpleasant and intrusive treatments like biopsies, which may be uncomfortable and have inherent dangers. This eliminates the need for patients to undergo these operations. Therefore, diagnostics that are based on exosomes provide a patient-friendly approach since they need nothing more than the collection of blood or urine [273].

A cutting-edge technique for detecting cancer is liquid biopsies. They include the noninvasive extraction and analysis of exosomes from body fluids, such as blood, urine, and saliva [274]. Exosomes are more practical and genuine than CTCs and circulating tumor DNA (ctDNA) [275]. An example of this would be the abundance of exosomes, which may be found in bodily fluids in quantities of around 109 particles per milliliter. This makes the collection of these vesicles much easier. However, Cai et al. [276] discovered that each milliliter of blood samples had very few CTCs. Additionally, living cells release exosomes, which contain biological data from the parent cells. However, exosomes are more representative of the biological information than cfDNA, which is released upon necrosis or death [276]. Exosomes, on the other hand, are fundamentally stable because of the lipid bilayer composition that they possess. This composition enables them to circulate consistently under physiological settings, even when they are subjected to the severe conditions that are present in the microenvironment of the tumor. This innate biological stability makes it possible to preserve samples for an extended period of time, which is necessary for the separation and identification of exosomes [277]. The analysis of exosomes is straightforward and uncomplicated. Particular proteins, including CD63, ALG-2-interacting protein X (ALIX), TS101, and HSP70,20, are expressed by exosomes. These proteins may be used as markers to efficiently differentiate exosomes from other vesicles. Exosomes are shown to be found in almost all of the body’s fluids and are characterized by their remarkable stability. They are encased in lipid bilayers. There is a similar quality of exosomal markers in samples that have been held at 4 °C for 24 h and subsequently at a temperature of 80 °C, samples that have been promptly stored at 80 °C, and fresh urine samples [278]. Liquid biopsies performed on exosomes have the potential to aid early cancer identification, maybe even before symptoms manifest themselves.

### 3.3. Limitations and Overcoming Limitations of Exosome Diagnostic Use in the Clinic

Exosomes must be quickly identified with few impurities and abnormalities for clinical use in order to guarantee purity. Despite being the most time-consuming technique, ultracentrifugation is still the gold standard for isolating and purifying exosomes. If exosomes are to become attractive for therapeutic usage, newer, faster techniques for separating them must be devised. These methods for clinical exosome identification and measurement have advantages and disadvantages. For example, enzyme-linked immunosorbent assay (ELISA) requires preparation time, while colorimetric response may provide a positive result rapidly when the test is finished. Although flow cytometry can identify different exosomes and classify them by species, it has drawbacks, including the requirement for costly equipment and variable findings. In addition to requiring costly equipment, nanoparticle tracking analysis (NTA) cannot distinguish between the several types of exosomes it discovers. New sensing and detecting techniques, such as electrochemical detection, microfluidics, and nanosensors, are presently being developed to address these limits.

Exosomes produced from tumors may be expensive and time-consuming to extract from bodily fluids. However, the usefulness of microfluidic chips and other nano-sensing technologies has been shown in a variety of point-of-care applications, such as cell sorting, fertility, DNA and protein sensing, virus detection, drug administration, cancer diagnostics, and tumor modeling [279].

Electrochemical sensors are among the most often used kinds of biosensors. These days, respiratory CO_2_ monitors and blood glucose monitors are common sensors. Antibodies are used by electrochemical-detecting sensors to interact with and collect antigens. Changes in the electric current influence the concentration of collected antigens [280,281]. Exosomes are a simple application of this idea. Numerous electrochemical tools have been developed to collect and examine tumor exosomes. Integrated magnetic-electrochemical exosomes, or iMEX, are one such device [281]. Included in IMEX are magnetic beads modified with the typical exosome tetraspanins CD9, CD63, and CD81. These exosomes that have been trapped by magnetic beads are drawn to the electrodes for electromagnetic detection by the system using magnets. To determine if the device may be useful in clinical settings, they used beads functionalized with the exosome indicator antibodies CD63, CA-125, EpCAM, and CD24, which are unique to ovarian cancer. This technique allowed iMEX to identify ovarian cancer exosomes in around an hour from 10 μL of patient blood [281]. To guarantee the repeatability of the assays, extra attention must be made in selecting the right markers and making sure the design is precise [280,282]. Nevertheless, these sensors might help exosome-based diagnostics get over its present problems.

The challenges encountered in the clinical use of exosomes for cancer monitoring and diagnostics may potentially be addressed by microfluidic devices. Microfluidic chips having many channels for sample isolation, washing, processing, and analysis may be made of silicon or plastic. However, their main disadvantage is that they usually need to be operated by knowledgeable personnel and may include many procedures with little automation [280]. In any case, they provide a potentially quick and affordable way to analyze tumor exosomes. Exosomes are distinguished from other extracellular vesicles, including CD63, MHCI, and CD83 by certain surface markers, according to earlier reports [192]. The device developed by Fang et al. uses CD63 and its antibody customized to magnetic beads. These beads are incubated for a whole night using cell culture media from breast cancer and ordinary cell lines or breast cancer and normal individual plasma [192]. Exosomes were combined with MHCI antibodies on the chip to separate certain tumor-associated exosomes, and they were then stained by immunofluorescent for HER2 and EpCAM [192]. Microfluidic technology has a promising clinical application prospect due to its small sample size (5–100 μL), high purity, high sensitivity, and short operation time (1.5 h), even though its use in exosome isolation is still in its infancy [283].

Using nanosensors is another method to overcome the limitations of exosome-based screening and diagnostics in the clinic. Nanosensors may also be relatively priced while retaining sensitivity since they often use a variety of methods, such as optics, electrochemical detection, and microfluidics [284,285]. Electrochemical sensing is included in the design of one tumor exosome detection nanodevice. This system employs platinum electrodes functionalized with EpCAM antibodies for tumor-exosome capture, and an additional EpCAM antibody tagged with a reporter is used for signal amplification [286]. Finding a tumor-specific exosome among the other circulating exosomes and healthy cells may be made easier with the capacity to identify single exosomes. This instrument would therefore be highly helpful in a medical setting. More research is required to assess this device using patient specimens, which contain tumor-specific antibodies like HER2 or EpCAM.

## 4. The Application of Exosomes in Clinical Trials

### 4.1. Trends of Trials Investigating Diagnostic and Prognostic Values of Exosomes

In the present study, we specify around five keywords, which include “Exosome”, “Exosome of mesenchymal stem cells (MSC)”, “EVs”, “Malignancy”, and “Non-malignancy”. This means that after searching with those keywords, we evaluated the title and content of trials and recognized which trial matches best with our aims and scope to identify relevant clinical trials. Out of 474 trials, 200 were removed since they were duplicates, and 274 others were excluded after assessing titles and content because they were against the backbones of the aims. Therefore, after a multi-step screening process, 118 relevant studies were included. Through a conceptual analysis of these trials, we aim to present a clearer understanding of the progress made so far and identify the areas that still need attention in the future.

Following the analysis of trials, we figured out that 21% of trials are in the completed status and around 6% acquire either terminated, withdrawn, or suspended status. Although 29% of trials are recruiting patients, 6% are active but do not recruit, and 8% neither recruit nor are active. These data demonstrate that major proportion of the trials in this context are completed or in the recruiting situation, showing that using diagnostic and prognostic values of exosomes in clinical trials has a growing pattern. Near 90% of clinical trials were designed to evaluate the features of exosomes in the adult population, while the relevant proportion for trials evaluating both child as well as child and adult simultaneously were 3% and 8%, respectively. Also, around 85% of trials considered both genders in their assessments, and the proportion for only-female and only-male trials were 8% and 7%, respectively. Overall, the number of all trials evaluating the diagnostic and prognostic features of exosomes has a rising pattern since 2012; the number of trials in malignancies have always been superior compared with non-malignant disorders. Furthermore, the world distribution shows that China, United States, and France are the countries where most of the studies have been conducted. Figure 1 shows an overview of trials in the context of diagnostic and prognostic values of exosomes. Additionally, the global distribution reveals that the majority of studies have been carried out in China, the United States, and France. Figure 1 shows an overview of trials in the context of diagnostic and prognostic values of exosomes.

### 4.2. Trend of Exosome Application in Malignancies Has Been Higher Compared with Non-Malignant Disorders

Generally, the proportion toward studies evaluating the diagnostic and prognostic capacity of exosomes in malignant disorders is 63% (74 trials), while it is 37% (44 trials) in non-malignant disorders. When we delved deeper into the analysis of malignancies, we figured out that GI cancers account for the most studies with 38% of cases, and after them, thoracic cancers (28%) and urogenital cancers (19%) are in the second and third levels. In the context of non-malignant disorders, cardiovascular disorders (16%), diabetes (14%), breathing disorders (11%), infections (9%), brain disorders (9%), and autoimmune diseases (9%) are the most studied diseases. It could be concluded that the variety of diseases in non-malignant disorders is higher; however, the absolute count of clinical trials in this category is lower compared with malignancies (Figure 2).

In another aspect of our analysis, we tried to shed light on the sources of derived exosomes and the nature of evaluated markers in exosomes. In this regard, we found that blood is the main source that exosome could be obtained, with 23% of cases. We separated other components of whole blood and figured out that after blood, plasma and serum were in the second and third levels with 23% and 18% of cases, respectively. Apart from blood and its components, which is the easiest and most valuable source, urine could serve as one of the best options (17%), particularly in disorders related to renal and urogenital disorders. All sources of exosomes, including blood, urine, tissue, CSF, umbilical cord, and so on, are provided in Figure 3. Regarding markers, proteins make up the largest share with 32% of cases, followed by miRNA (27%) and RNA (12%). It is obvious that studies mostly concentrate on proteins and miRNAs, as these biomolecules not only alter in disorders but also are able to modulate the situation of diseases; therefore, their application is popular (Figure 3). Table 1 also provides a full list of clinical trials in the context of exosomes and their roles as diagnostic and prognostic tools in malignant and non-malignant diseases.

## 5. Conclusions and Future Prospect

Exosomes have emerged as pivotal elements in the realm of diagnostics, reflecting their essential role in intercellular communication and biomarker potential. With their ability to carry and preserve a diverse array of molecular contents such as proteins, lipids, and nucleic acid, exosomes represent a real-time and noninvasive diagnostic tool. Their growing significance is underscored by their use in identifying and monitoring conditions from malignancies to non-malignant disorders like cardiovascular diseases and neurodegenerative disorders.

We demonstrated that exosomes could serve as diagnostic and prognostic biomarkers, particularly in cancer, where their unique molecular cargo provides insights into tumor progression. Indeed, exosomes are a new source of biomarkers capable of serving as diagnostic and prognostic tools. As they are stable, they could maintain the cargoes related to a specific type of disease or a pathologic disorder more constant, and that is why they are potent specific biomarkers. Furthermore, their count could correlate with the burden of disorders such as cancers; therefore, exosome quantification can be a noninvasive method for cancer monitoring and prognosis [287]. Studies have emphasized the diagnostic and prognostic value of quantifying plasmatic exosome levels in cancer patients. Indeed, the exosome levels are significantly higher in tumor patients and decrease after tumor removal. The exosome count could be a potential addition to routine blood tests, as it could be measured easily. Standardizing exosome detection and quantification methods could be one of the future areas in diagnostic and prognostic studies [288]. All in all, it could be inferred that the count of exosomes or tumor antigen-expressing ones could be another advantage of exosomes in the diagnostic and prognostic assessments, as they can be effortlessly measured and are associated with the development of various tumors such as melanoma and prostate cancer. According to our analysis, the number of clinical trials conducted in the context of malignancies exceeds the non-malignant ones, which could be due to the success of exosome-based diagnostic and prognostic tools in cancer-related studies. The most utilized forms of biomolecules in clinical trials have been proteins and non-coding RNAs. Exosomal proteins and miRNAs have demonstrated remarkable specificity and sensitivity in distinguishing pathological states, such as differentiating malignant from benign conditions in gastrointestinal, thoracic, and urogenital tumors. The application of exosomes as tools to facilitate the diagnosis and predict the prognosis of cancers has been mainly focused on GI, thoracic, and urogenital tumors, while cardiovascular disorders, diabetes, and breathing disorders have drawn the most attention in the context of non-malignant diseases, reinforcing their relevance in a clinical context.

As exosomes could serve as the representer of cells, they are highly specific to different conditions, tissues, and cell types; therefore, their alterations, whether in their numbers or their contents, could be associated with certain diseases, particularly cancers. Furthermore, they are mainly collected by noninvasive approaches, meaning that utilizing their diagnostic and prognostic values is patient-friendly. It is worth mentioning that exosomes are found in a wide range of body fluids and are recognized for their remarkable stability. Nonetheless, their clinical utility is still being hampered by challenges related to standardization of isolation techniques, marker identification, and scalability. It has been demonstrated that electrochemical sensors, such as iMEX, are studied for their ability to detect tumor exosomes using magnetic beads. Microfluidic devices offer a fast and cost-effective solution to isolate and analyze exosomes with minimal sample volumes. Nanosensors also improve sensitivity, incorporating electrochemical and optical approaches to detect tumor-specific exosomes at very low concentrations.

It could be inferred that advancements and innovations such as integrating exosome markers with artificial intelligence models might facilitate diagnostic workflows, leading to a more accurate evaluation of complex disorders. Moreover, combination strategies that use multi-omics exosomal profiling could potentially unravel intricate disease mechanisms, advancing personalized medicine. Resolving the current technical challenges could unleash the full potential of exosomes as capable diagnostic and prognostic tools in modern healthcare.

## Figures and Tables

**Figure 1 biomolecules-15-00587-f001:**
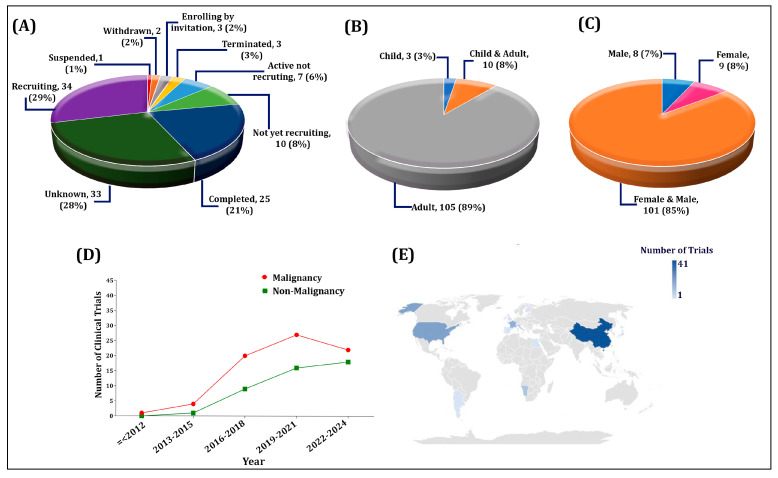
Diagnostic and prognostic values of exosomes in clinical trials. An overview of exosome application in clinical trials is provided in this figure. (**A**) Status of clinical trials. (**B**) Age of volunteers. (**C**) Gender of volunteers. (**D**) Trend of clinical trial numbers since 2012. (**E**) World distribution of clinical trials.

**Figure 2 biomolecules-15-00587-f002:**
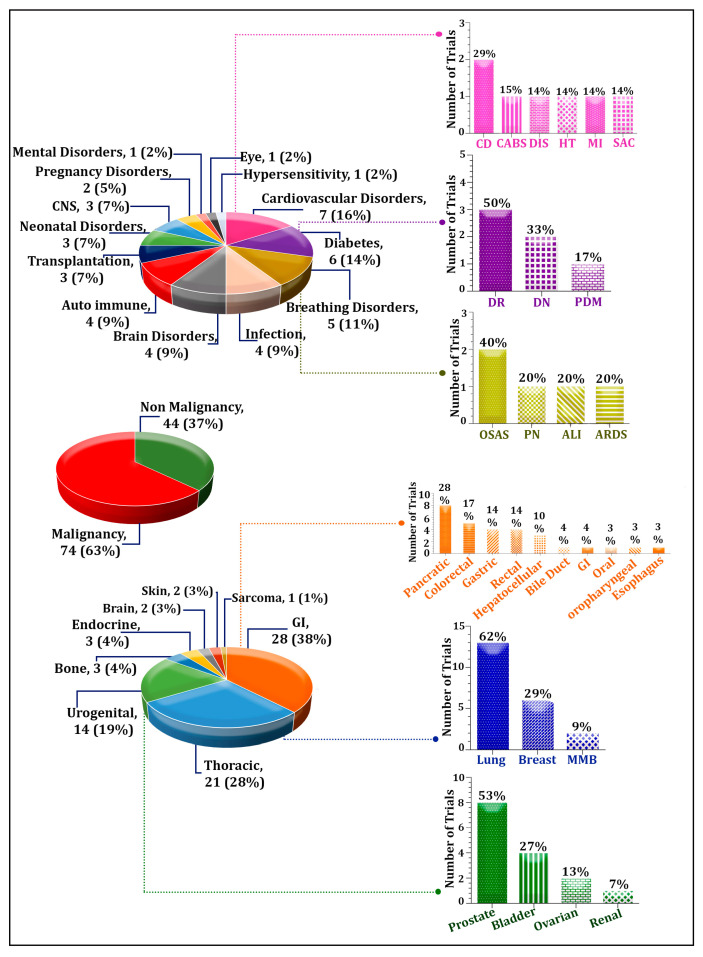
Details of disorders in which the diagnostic and prognostic values of exosomes have been assessed. Generally, the diseases could be categorized into malignant and non-malignant disorders, with malignancies possessing the majority of cases (63%). In the non-malignant disorder category, cardiovascular disorders, diabetes, and breathing disorders have accounted for the most cases, respectively. In the malignancy category, gastrointestinal (GI), thoracic, and urogenital cancers have been, in order, the most cases. Cardiovascular disease (CD), coronary artery bypass surgery (CABS), delirium in cardiovascular surgery (DIS), hypertension (HT), myocardial infarction (MI), stratification of adverse cardiac (SAC), diabetic retinopathy (DR), diabetic nephropathy (DN), post-pancreatitis diabetes mellitus (PDM), obstructive sleep apnea syndrome (OSAS), pulmonary nodules (PN), acute lung injury (ALI), acute respiratory distress syndrome (ARDS), gastrointestinal (GI), metastatic meningitis from breast (MMB).

**Figure 3 biomolecules-15-00587-f003:**
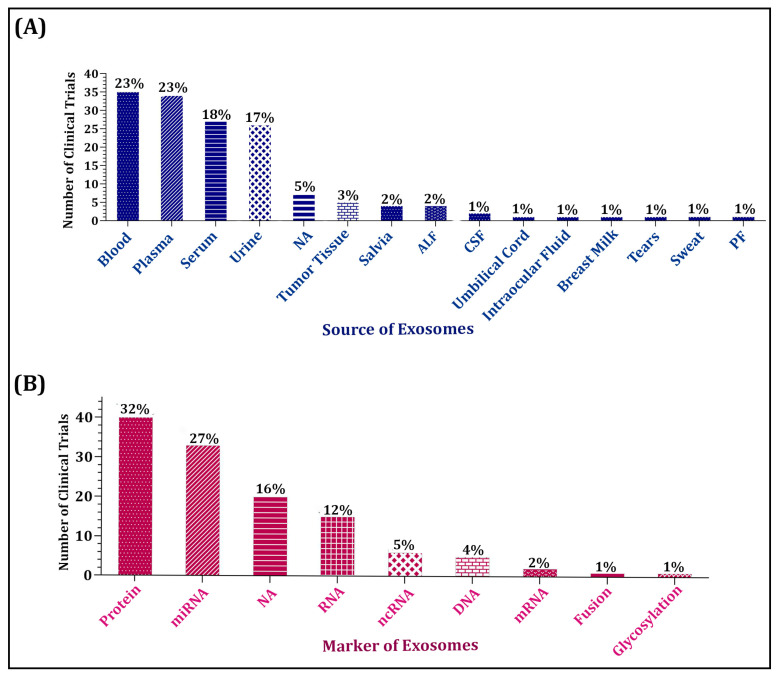
The sources of exosome collection and the markers expressed in exosomes. (**A**) The most utilized sources in clinical trials have been blood and blood derivates, such as plasma and serum, along with urine. The other sources possess a smaller proportion. (**B**) Proteins and miRNAs have been the most evaluated markers, which could have diagnostic and prognostic features. Alveolar lavage fluid (ALF), cerebrospinal fluid (CSF), pancreatic fluid (PF), Not Available (NA).

**Table 1 biomolecules-15-00587-t001:** The characteristics of Completed clinical trials (n = 25 trials).

No.	NCT Number	St. Date	Sex	Age	Enrollment	Disease Type	Source of Exosome	Associated Marker
Malignancy								
GI Cancer								
1	NCT06342427	2023	ALL	Adult	809	Gastric Cancer	Serum	miRNA
2	NCT06023121	2018	ALL	Adult	800	Gastric Cancer	Blood	LncRNA
3	NCT06469892	2020	ALL	Adult	225	Oral Cancer	Plasma & Saliva	miR-185
4	NCT03032913	2017	ALL	Adult	52	Pancreatic Cancer	Plasma	NA
5	NCT04394572	2021	ALL	Adult	80	Colorectal Cancer	Serum	Protein
Thoracic Cancer								
6	NCT02890849	2016	ALL	All	60	Lung Cancer	Plasma	mRNA
7	NCT03830619	2017	ALL	Adult	1000	Lung Cancer	Serum	lncRNA
8	NCT02869685	2017	ALL	Adult	60	Lung Cancer	Plasma	miRNA
9	NCT03228277	2017	ALL	Adult	25	Lung Cancer	Serum	DNA
Urogenital Cancer								
10	NCT02702856	2014	MALE	Adult	2000	Prostate Cancer	Urine	RNA
11	NCT04720599	2020	MALE	Adult	120	Prostate Cancer	NA	NA
12	NCT06193941	2023	ALL	Adult	400	Bladder Cancer/Urothelial Carcinoma	Urine	RNA
Bone Cancer								
13	NCT05101655	2020	ALL	Child	60	Osteosarcoma Lung Recurrence	Plasma	Protein
14	NCT03895216	2018	ALL	Adult	34	Bone Metastases	Plasma	miRNA
15	NCT03488134	2018	ALL	Adult	74	Thyroid Cancer	Urine	Protein
Non-Malignancy								
Cardiovascular Disorders								
1	NCT03034265	2016	ALL	Adult	24	Hypertension	Urine	Protein
2	NCT02226055	2014	ALL	Adult	200	Cardiovascular	Serum & Urine	NA
Breathing disorders								
3	NCT03811600	2019	ALL	Adult	90	Obstructive Sleep Apnea Syndrome	Plasma & Serum	Protein
4	NCT04459182	2021	ALL	Adult	99	Obstructive Sleep Apneas Hypopneas Syndrome	NA	miRNA
Autoimmune Disorder								
5	NCT03984006	2019	ALL	Adult	5	Autoimmune Thyroid Heart Disease	Urine	Protein
Brain Disorder								
6	NCT03419000	2018	ALL	Adult	75	Drug-Resistant Epilepsy	Blood	miRNA
CNS Disorder								
7	NCT01860118	2023	ALL	Adult	601	Parkinson’s Disease	Blood & Urine	Protein
Infection								
8	NCT03267160	2017	ALL	Adult	30	Hemodynamic Instability	Blood & Urine	Protein
Pregnancy disorder								
9	NCT03562715	2016	FEMALE	Adult	200	Preeclampsia	Umbilical Cord	miRNA
Transplantation								
10	NCT03503461	2018	ALL	Adult	67	Kidney Transplantation	Urine	Protein

GI: gastrointestinal; miRNA: MicroRNA; LncRNA: long non-coding RNA; NA: Not Available; mRNA: messenger RNA; CNS: central nervous system.

## Data Availability

Data sharing is not applicable to this article as no datasets were generated or analyzed during the current study.

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
