# Peer review of "Exosomal Biomarkers: A Comprehensive Overview of Diagnostic and Prognostic Applications in Malignant and Non-Malignant Disorders"

_biomolecules, 2025, doi:10.3390/biom15040587_

Round 1

Reviewer 1 Report

Comments and Suggestions for Authors

This is a very comprehensive overview of the diagnostic and prognostic application of exosomes in malignant and non-malignant disorders. The review covers many areas of potential clinical application of exosomes in liquid biopsies.   However, there are many limitations that the authors noted that somehow dampens the interest of the reader.  It appears as if the exosomes they cited in their studies were isolated and purified using different techniques that may impact the protein as well as the microRNAs composition of the exosomes in the numerous studies they cited.  For example, exosomes isolated using polymer precipitation technique such as ExoQuick may yield exosomes with a different repertoire of proteins compared to a study in which differential centrifugation techniques are used using the same source such as urine or saliva.  It would have been more consistent if all the exosomes examined were isolated and purified using the same technique.  Apart from these limitations, the authors should also consider the following in their revisions.

  • There are numerous repetitions in the manuscript that need to be corrected. For example, in lines 54-92, the authors compared the potential advantage of using exosomes for diagnosis and prognosis to traditional solid biopsies more than once.
  • The authors stated that exosomes are extremely stable but did not provide a reference (line 170).
  • At line 213, overexpression of miRNAs such as miR-320a, miR-22, miR-423-5p and miR-92b in serum exosomes has been associated with heart failure but no reference was provided.
  • In line 1159, comparison of diagnostic and prognostic capacity of exosomes in malignant disorders is compared with what? This is not clear.
  • Compared to exosomal proteins, it appears from their review that exosomal miRNAs are better diagnostic and prognostic biomarkers in malignant disorders. I suggest that they make another table where they list the miRNAs that are associated with all the malignant disorders and also provide references.
  • Lines 1125 to 1134 is a duplication of lines 1114 to 1124.
  • In line 872, the authors mentioned the …the levels of these miRNAs in exosomes. Which miRNAs??  Also provide reference.
  • Table 3A-…Saliva not Salvia
Comments on the Quality of English Language

There were many points that the authors could have used simple phrases.  Used many words to explain a simple point. The review could be compressed into half of the pages used. 

Author Response

We would like to thank you for the opportunity to resubmit a revised copy of the manuscript biomolecules-3512362 entitled “Exosomal Biomarkers: A Comprehensive Overview of Diagnostic and Prognostic Applications in Malignant and Non-Malignant Disorders”. We also appreciate the positive feedback and helpful comments of the reviewers. The manuscript has been revised to address the comments, which are appended alongside our responses to this letter.

Sincerely yours,

Davood Bashash, Ph.D.

Associate Professor of Hematology

Reviewer #1:

This is a very comprehensive overview of the diagnostic and prognostic application of exosomes in malignant and non-malignant disorders. The review covers many areas of potential clinical application of exosomes in liquid biopsies. However, there are many limitations that the authors noted that somehow dampens the interest of the reader. It appears as if the exosomes they cited in their studies were isolated and purified using different techniques that may impact the protein as well as the microRNAs composition of the exosomes in the numerous studies they cited.  For example, exosomes isolated using polymer precipitation technique such as ExoQuick may yield exosomes with a different repertoire of proteins compared to a study in which differential centrifugation techniques are used using the same source such as urine or saliva. It would have been more consistent if all the exosomes examined were isolated and purified using the same technique. Apart from these limitations, the authors should also consider the following in their revisions.

  • Thank you for your kind and positive feedback. We truly appreciate your time and help. Indeed, the purpose of the study was to provide a comprehensive overview of diagnostic and prognostic values of exosomes. Therefore, we tried to cover all aspect of exosomes like the way they could be purified or the cargoes they could contain. We agree with your opinion that this could influence the efficacy of exosomes; thus, we provided a section to reflect the challenges and shortcomings, since we believe in truthfulness in expressing scientific information.

  1. There are numerous repetitions in the manuscript that need to be corrected. For example, in lines 54-92, the authors compared the potential advantage of using exosomes for diagnosis and prognosis to traditional solid biopsies more than once.
  • Thank you for pointing this out. We carefully checked the whole manuscript and corrected the repetition sections to provide a more integrated text. The revised mentioned sections are as follows:
    • A sequence of imaging scans is used to diagnose different tumor forms, and a biopsy is then performed to verify the diagnosis. In addition to being intrusive, these techniques may be expensive and painful for the patient. Additionally, individuals are more probable to acquire a late-stage diagnosis if regular screening is not performed. One possible disadvantage of a delayed diagnosis is that it may reduce the patient's reaction to treatment and, therefore, their chances of survival. Because of this, it's critical to track how well a patient is responding to treatment. Many of traditional cancer detection methods are inappropriate for patients because they need intrusive procedures, such as surgeries, to collect tissue samples for examination. These operations may be risky and uncomfortable. Furthermore, they may be expensive, burdening patients and healthcare systems alike, especially when many tests are needed to verify a diagnosis or track the effectiveness of therapy. As a consequence, smaller tumors or malignant cells may go undetected until they have progressed to more severe and un-treatable phases. A growing effort has been made to develop less invasive, cost-effective, highly sensitive diagnostic techniques, including liquid biopsies and sophisticated imaging technologies, that have the potential to detect tumors early and monitor patients undergoing therapy in order to conquer these limitations. By providing earlier detection, less invasiveness, and more accurate monitoring, these advancements have the potential to completely transform the diagnosis of cancer and strengthen our fight against this terrible illness.

In the area of cancer detection, exosomes have become a rapidly developing re-search area with enormous promise. Exosomes are also essential for a number of pathological as well as physiological processes such as immunological responses, cancer, pregnancy issues, and cardiovascular problems. Exosomes were discovered to be present in almost every bodily fluid, including blood, urine, Cerebrospinal fluid (CSF), saliva, pleural effusion, ascites fluid, amniotic fluid, breast milk, and bronchoalveolar lavage fluid (BALF), according to the many investigations that followed. The pilot study evaluated saliva collection methods for exosome isolation and characterized exosomal proteins in cancer-free controls, oral potentially malignant disease (OPMD), and oral squamous cell carcinoma (OSCC) patients. A three-protein panel (AMER3, LOXL2, AL9A1) distinguished cancer-free individuals from OPMD/OSCC with high accuracy (AUC 0.93). Another panel (PSB7, AMER3, LOXL2) showed strong diagnostic potential for OSCC (AUC 0.96), suggesting its promise as a biomarker for early detection. Another study found significant differences in the size, concentration, and biomolecular composition of salivary exosomes between healthy individuals, tobacco consumers, and oral cancer patients. Tobacco users and cancer patients had larger and more concentrated exosomes, suggesting early cellular changes. These findings highlight the potential of salivary exosomes as noninvasive biomarkers for distinguishing high-risk individuals and enabling early oral cancer detection. Sweat is a simple and less interference-prone biofluid, making it valuable for disease biomarker studies. Sweat exosome-based detection shows potential for lung cancer screening through molecular expression analysis. Notably, Dermcidin (DCD) and Prolactin Inducible Protein (PIP) serve as biomarkers, with DCD linked to breast cancer and lymph nodes, while PIP is overexpressed in breast and prostate cancer. Tear exosomes contain higher levels of exosome markers (CD9, CD63) than serum exosomes and were analyzed using qRT-PCR and western blot. Breast cancer-specific miR-21 and miR-200c were highly expressed in tear exosomes from meta-static breast cancer patients compared to healthy individuals (36). Cholangiocarcinoma (CCA) is an aggressive cancer with poor prognosis and limited diagnostic or treatment options. This study identified two exosomal lncRNAs, ENST00000588480.1 and ENST00000517758.1, with significantly increased expression in bile from CCA patients. Their combined diagnostic performance (AUC: 0.709, sensitivity: 82.9%, specificity: 58.9%) correlated with cancer progression and poor survival. KEGG analysis highlighted their involvement in the p53 signaling pathway, suggesting their potential as non-invasive bile-based biomarkers and therapeutic targets for CCA.

  1. The authors stated that exosomes are extremely stable but did not provide a reference (line 170).
  • As recommended, the section was revised as follows:
    • Exosomes have slower rate of degradation and are extremely stable in stored patient samples. Indeed, as exosomes are membranous structures, the contents could be ade-quately protected from degradation by extracellular proteases and can be highly stable in storage conditions. This makes the application of exosomes in clinical settings possible because samples may be stored for extended periods of time before being analyzed. Due to the fact that exosomes exhibit surface markers that are associated with their cellular origin, it is possible to trace them back to their point of origin. Nonetheless, the stability of exosomes could be influenced by storing conditions. In this regard, the biodistribution and uptake efficiency could be notably reduced following storage at 4 °C and −20 °C. Exosomes, however, maintain their biodistribution at −80 °C for 14 days. Also, it was shown that the number of exosomes, their size, their capacity to express protein at the surface, and functional stability could be constant at −70 °C after 25 days.
  1. At line 213, overexpression of miRNAs such as miR-320a, miR-22, miR-423-5p and miR-92b in serum exosomes has been associated with heart failure but no reference was provided.
  • As recommended, the section was revised and relevant references were added as follows:
    • Overexpression of microRNAs miR-22, miR-320a, miR-423-5p, and miR-92b in serum and serum exosomes has been associated with heart failure (HF). For the purpose of diagnosing and determining the prognosis of systolic heart failure, these biomarkers can be utilized as specific indicators. Patients with heart failure had higher levels of many serum-based exosomes encoding p53-responsive microRNAs after one year had passed since the beginning of acute myocardial infarction. These miRNAs included miR-34a, miR-192, and miR-194. Additionally, it was observed that the expression levels of exosomes miR-194 and miR-34a, but not miR-192, were strongly connected with left ventricular diastolic size and left ventricular ejection percentage, when tested about one year following the beginning of acute myocardial infarction (AMI).
  1. In line 1159, comparison of diagnostic and prognostic capacity of exosomes in malignant disorders is compared with what? This is not clear.
  • Thank you for pointing this out. The section was edited as follows:
    • 2. Trend of exosome application in malignancies has been higher compared with non-malignant disorders

Generally, the proportion towards studies evaluating the diagnostic and prognostic capacity of exosomes in malignant disorders is 63% (74 trials), while it is 37% (44 trials) in non-malignant disorders.

  1. Compared to exosomal proteins, it appears from their review that exosomal miRNAs are better diagnostic and prognostic biomarkers in malignant disorders. I suggest that they make another table where they list the miRNAs that are associated with all the malignant disorders and also provide references.
  • Thank you for the recommendation. The details of utilized miRNAs along with proteins, lncRNAs and other sources were provided in Table S1. Since the numbers of studies were notably high, we decided to transfer them into the supplementary data. However, because we value your comment, we provided a separated table regarding miRNAs as Table S1 and the rest data were transferred to Table S2. The new Table S1 is as follows:

Table S1. Studies using miRNAs as the diagnostic and prognostic tools in exosomes.

Category

No.

miRNAs

Sample

Expression

Application

Ref.

Brain Cancer

1

miR-29c-3p and miR-219-5p

Serum

Decrease

Prognosis

[1]

2

miR-190a

Serum

Increase

Prognosis

[1]

3

miR-451, miR-711, and miR-935,

CSF

Increase

Diagnosis

[2]

4

miR-21

Serum

Increase

Diagnosis

[3]

5

miR-182

Tissue

Increase

Prognosis

[4]

6

miR-486

Tissue

Increase

prognosis

[5]

7

miR-328

Tissue

Decrease

Diagnosis and Prognosis

[6]

8

miR-210

Serum

Increase

Diagnosis and Prognosis

[7]

9

miR-2276-5p

Plasma

Decrease

Diagnosis and Prognosis

[8]

10

miRNA-19a and miRNA-19b

Serum

Decrease

Diagnosis

[9]

11

miR-155-5p

Plasma

Increase

Diagnosis and Prognosis

[10]

12

miR-98-5p, miR-323-3p, miR-19b-3p

Serum

Decrease

Diagnosis

[11]

13

miR-183-5p

Serum

Increase

Diagnosis

[11]

14

miR-9-5, miR-21-5p, miR-1-3p

Tissue and Cell lines

Increase

Prognosis

[12]

15

miR-138-5p

Tissue and Cell lines

Decrease

Diagnosis and Prognosis

[12]

16

miR-29b

Serum

Decrease

Diagnosis and Prognosis

[13]

17

miR-454-3p

Serum

Increase

Diagnosis and Prognosis

[14]

18

miR-21, miR-222 and miR-124-3p

Serum

Increase

Prognosis

[15]

Lung tumor

1

miR-451a

Plasma

Increase

Prognosis

[16]

2

miR-96

Serum

Increase

Diagnosis and Prognosis

[17]

3

miR-19b-3p, miR-21-5p, miR-221-3p, miR-409-3p, miR-425-5p and miR-584-5p

Plasma

Increase

Diagnosis

[18]

4

miR-126

Serum

Decrease

Diagnosis

[19]

5

let-7a and miR-155

Serum

Decrease

Diagnosis

[20]

6

miR-21

Serum

Decrease

Diagnosis

[21]

7

let-7f, miR-126-3p, miR-148b, miR-151-5p, miR-199a-3p, miR-221, miR-23b, miR-26a, miR-27b, and miR-423-3p

Serum

Increase

Diagnosis

[22]

8

miR-23b-3p, miR- 10b-5p, miR-21- 5p

Plasma

Increase

Prognosis

 [23]

Breast Cancer

1

miR-21 and mi-R1246

Plasma

Increase

Diagnosis

[24]

2

miR-145, miR-155 and miR-382

Serum

Increase

Diagnosis

[25]

3

miR-223-3p

Plasma

Increase

Diagnosis

[26]

4

miR-16, miR-93

Plasma

Increase

Diagnosis

[27]

5

miR-30

Plasma

Decrease

Prognosis

[27]

6

miR-188-3p, miR-500a-5p and miR-502-3p

(miR-532-502 cluster)

Serum

Increase

Diagnosis

[28]

7

let‐7b‐5p, miR‐122‐5p, miR‐151a‐3p, miR‐215‐5p, miR‐223‐5p, miR‐23a‐3p, miR‐660‐5p, miR‐126‐5p, miR‐146b‐5p, miR‐210‐3p and miR‐222‐3p

Plasma

Increase

Diagnosis

[29]

8

miR-106a-3p, miR-106a-5p, miR-20b-5p, and miR-92a-2-5p

Plasma

Increase

Diagnosis

[30]

9

miR-106a-5p, miR-19b-3p, miR-20b-5p, and miR-92a-3p

Serum

Increase

Diagnosis

[30]

10

miR-16-5p, miR-106a-5p, miR-25-3p, miR-425-5p, and miR-93-5p, miR-20a-5p and miR-223-3p

Tissue

Increase

Diagnosis

[31]

11

let-7b-5p

Tissue

Decrease

Diagnosis

[31]

12

miR-148a

Serum

Decrease

Prognosis

[32]

13

miR-21 and miR-105

Serum

Increase

Diagnosis and Prognosis

[33]

Pancreatic Cancer

1

miR-155 and miR-196a

Serum

Decrease

Diagnosis

[34]

2

miR-17-5p

Serum

Increase

Diagnosis

[34]

3

miR-214

Tissue

Decrease

Prognosis

[35]

4

miRNA-21

Serum and Pancreatic juice

Increase

Diagnosis and Prognosis

[36]

5

miRNA-21 and miRNA-210

Serum

Increase

Diagnosis and Prognosis

[37]

6

miR-10b, miR-21, miR-30c, and miR-181a

Serum

Increase

Diagnosis

[38]

7

miR-let7

Serum

Decrease

Diagnosis

[38]

8

miR-1246, miR-4644, miR-3976 and miR-4306

Serum

Increase

Diagnosis

[39]

9

miR-222

Plasma

Increase

Diagnosis and Prognosis

[40]

10

miR-4525, miR-451a and miR-21,

portal vein blood and PB

Increase

Diagnosis and Prognosis

[41]

11

miRNA-155

Pancreatic juice

Increase

Diagnosis and Prognosis

[42]

12

miR‑1246, miR‑3976, miR‑4306, and miR‑4644

Saliva and Serum

Increase

Diagnosis

[43]

Colorectal Cancer

1

miR-23a and miR-301a

Serum

Increase

Diagnosis

[44]

2

miR-193a

Plasma

Decrease

Prognosis

[45]

3

let-7g

Plasma

Increase

Prognosis

[45]

4

miR-548c-5p

Serum

Decrease

Diagnosis

[46]

5

miR-92b

Plasma

Decrease

Diagnosis

[47]

6

miR-27a-5p

Serum

Decrease

Diagnosis

[48]

7

miR-224-5p, miR-548d-5p, miR-200a-3p, miR-320d, miR-200b-3p, and miR-1246

Plasma

Increase

Diagnosis

[48]

8

miR-6803-5p

Serum

Increase

Diagnosis and Prognosis

[49]

9

miR-27a and miR-130a

Plasma

Increase

Diagnosis

[50]

10

miR-122

Serum and cell line

Increase

Diagnosis

[51]

11

miR-548c-5p

Serum

Decrease

Prognosis

[51]

12

miR-99b-5p and miR-150-5p

Serum

Decrease

Diagnosis

[52]

Hepatocellular carcinoma

1

miR-122, miR-125b, miR-145, miR-192, miR-194, miR-29a, miR-17-5p, miR-106a

Serum

Increase

Diagnosis

[53]

2

miR-370-3p

Serum

Decrease

Diagnosis and Prognosis

[54]

3

miR -196a-5p

Serum

Increase

Diagnosis and Prognosis

[54]

4

miR-122, miR-148a, and miR-1246

Serum

Increase

screening

[55]

5

miR-21-5p, miR-10b-5p, miR-221-3p, miR-223-3p

Serum

Increase

Diagnosis

[56]

6

miR-18a, miR-27a and miR-20b

Plasma

Increase

Prognosis

[57]

7

miR-125b

Serum

Decrease

Diagnosis and Prognosis

[58]

8

miR-29a, miR-29c, miR-133a, miR-143, miR-145, miR 192, and miR-505

Serum

Increase

Diagnosis and Prognosis

[59]

9

miR-21-5p and miR-144-3p

Serum

Increase

Diagnosis

[60]

10

miR-21-5p

Serum

Increase

Diagnosis

[61]

11

miR-92a-3p

Plasma

Increase

Prognosis

[62]

12

miR-4661-5p

Serum

Increase

Diagnosis

[63]

13

miR-10b-5p, miR-18a-5p, miR-215-5p, and miR-940

Serum and cell line

Increase

Diagnosis and Prognosis

[64]

14

miR-483-5p

Plasma and tissue

Increase

Diagnosis

[65]

15

miR-101 and miR-125b

Serum

Decrease

Diagnosis

[66]

16

RNA-224

Serum

Increase

Diagnosis and Prognosis

[67]

17

miR665

Serum and tissue

Increase

Diagnosis and Prognosis

[68]

18

miR-18a, miR-221, miR-222, miR-224

Serum

Increase

Diagnosis

[69]

19

miR-101, miR-106b, miR-122, miR-195

Serum

Decrease

Diagnosis

[69]

20

miR-122 and miR-148a

Serum

Decrease

Diagnosis and Prognosis

[70]

21

miR-21

Serum

Increase

Prognosis

[71]

22

miR-92b

Serum

Increase

Prognosis

[72]

23

miR-320d

Serum

Decrease

Diagnosis and Prognosis

[73]

24

miR-638

Serum

Decrease

Prognosis

[74]

25

miR-125b

Serum

Decrease

Prognosis

[75]

Thyroid cancer

1

miR‑146b, miR-222

Cell line

Increase

Diagnosis

[76]

2

miR‑21‑5p, miR‑181a

Plasma

Increase

Diagnosis

[77]

3

miR‑346, miR‑10a‑5p, miR‑34a‑5p

Plasma

Increase

Diagnosis

[78]

4

miR-376a-3p, miR-4306, miR-4433a-5p, and miR-485-3p

Plasma and serum

Increase

Diagnosis and Prognosis

[79]

5

miR-16-2-3p, miR-223-5p, miR-34c-5p, miR-182-5p, miR-223-3p, and miR-146b-5p

Plasma

Decrease

Diagnosis

[80]

6

miR-16-2-3p and miR-223-5p

Plasma

Increase

Diagnosis

[80]

Prostate cancer

1

miR-21, miR-574, miR-375

Serum

Increase

Diagnosis

[81]

2

miR-200c-3p and miR-21-5p

Plasma

Increase

Diagnosis

[82]

3

miR-574-3p, miR-141-5p, and miR-21-5p

Urine

Increase

Diagnosis and Prognosis

[83]

4

miR-214

Urine

Decrease

Diagnosis

[84]

5

miR-21, miR-141, and miR-375

Urine

Increase

Diagnosis

[84]

6

miR-21 and miR-200c

Urine

Decrease

Diagnosis

[85]

7

miR-21, miR-204, miR-375

Urine

Increase

Diagnosis and Prognosis

[86]

8

miR-141

Serum

Increase

Diagnosis

[87]

9

miR-107 and miR-574-3p

Plasma and Urine

Increase

Diagnosis

[88]

10

miR-2909

Urine

Increase

Diagnosis

[89]

11

miR-196a-5p, miR-34a-5p, miR-143-3p, miR-501-3p and miR-92a-1-5p

Urine

Decrease

Diagnosis

[90]

12

miR-30b miR-126

Urine

Increase

Diagnosis

[91]

13

miR-1246

Serum

Decrease

Diagnosis

[92]

Kidney cancer

1

miR-141 and miR-200b

Supernatant

Increase

Prognosis

[93]

2

miR-210 and miR-1233

Serum

Increase

Diagnosis

[94]

3

miR-204-5p

Urine

Increase

Diagnosis

[95]

4

miR-92a-1-5p,

Plasma

Decrease

Diagnosis

[96]

5

miR-149-3p, miR-424-3p

Plasma

Increase

Diagnosis

[96]

6

miR-15a

Plasma

Increase

Diagnosis

[97]

7

miR-let-7i-5p, miR-26a-1-3p,

miR-615-3p

Plasma

Decrease

Prognosis

[98]

8

miR-210

Serum

Increase

Diagnosis

[99]

9

miR-224

Serum

Increase

Prognosis

[100]

Ovarian cancer

1

miR-222-3p

Serum

Increase

Diagnosis

[101]

2

miR-200a-3p, miR-766-3p, miR-26a-5p, miR-142-3p,

let-7d-5p, miR-130b-3p, miR-374a-5p, and miR-

328-3p

Serum

Increase

Diagnosis

[102]

3

miR-21, miR-141, miR-200a, miR-200b, miR-200c,

miR-203, miR-205, and miR-214

Serum

Increase

Diagnosis

[103]

4

miR-1307 and miR-375

Serum

Increase

Diagnosis

[104]

5

miR-34a

Serum

Increase

Diagnosis and Prognosis

[105]

6

miR-16, miR-93, miR-126, miR-223

Serum

Decrease

Prognosis

[106]

7

miR-21, miR-100, miR-200b, and miR-320

Serum

Increase

Prognosis

[106]

8

miR-146b-5p

Serum

Increase

Prognosis

[107]

9

miR-93, miR-145 and miR-200c

Serum

Increase

Diagnosis

[108]

10

miR-373 miR-200amiR-200bmiR-200c

Serum

Increase

Diagnosis and Prognosis

[109]

11

miR-30a-5p

Urine

Increase

Diagnosis and Prognosis

[110]

12

miRNA-205

Plasma

Increase

Diagnosis and Prognosis

[111]

13

miR-4732-5p, miR-877-5p, miR-574-3p, let-7a-5p, let-7b-5p, let-7c-5p, and let-7f-5p

Plasma

Increase

Diagnosis

[112]

14

miR-1273f, miR-342-3p

Plasma

Decrease

Diagnosis

[112]

15

miR-1290

Serum

Increase

Diagnosis

[113]

16

miR-1260a, miR-7977, miR-192-5p

Plasma

Decrease

Diagnosis and Prognosis

[114]

Cervical cancer

1

miR‐221‐3p

Cell line

Increase

Diagnosis

[115]

2

miR‐30d‐5p, let‐7d‐3p

Plasma

Decrease

Diagnosis

[116]

3

miR‐196a, miR‐486‐5p

Plasma

Increase

Diagnosis and Prognosis

[117]

4

miR‐125a‐5p

Plasma

Decrease

Diagnosis and Prognosis

[117]

5

miR-221, miR-222

Tissue

Increase

Diagnosis

[118]

6

miR‐21, miR‐146a

cervicovaginal lavage

Increase

Diagnosis

[119]

7

miR-10b-5p, miR-34b-3p, miR-34c-5p, miR-34c-3p, miR-449b-5p, miR-200b-3p, miR-383-5p, miR-2110

Peritoneal lavage

Decrease

Diagnosis

[120]

8

miR-125a-5p

Plasma

Decrease

Diagnosis

[121]

9

miR‐146a‐5p, miR‐15s1a‐3p, miR‐2110, miR‐21‐5p

Plasma

Increase

Diagnosis

[122]

10

miRNA‐20a, miRNA‐203, miRNA‐21, miRNA‐205, miRNA‐218, miR‐485‐5

NA

Increase

Diagnosis

[123]

11

miR‐877‐3p

Tissue

Increase

Diagnosis

[124]

12

miR‐24, miR‐451, miR‐125a

NA

Decrease

Diagnosis

[125]

13

miR‐130a

Cell line

Increase

Diagnosis

  [126]

14

miR‐155

Peripheral blood and tissues

Increase

Diagnosis

[127]

15

hsa-miR-200c-3p

Urine

Increase

Diagnosis

[128]

Melanoma

1

miR-15b-5p, miR-149-3p, miR-150-5p

Plasma

Increase

Diagnosis

[129]

2

miR-193a-3p, miR-524-5p

Plasma

Decrease

Diagnosis

[129]

3

miR-211-5p, miR-16

Serum

Increase

Diagnosis and Prognosis

[130]

4

miR-4487

Serum

Decrease

Diagnosis and Prognosis

[130]

5

miR-221

Serum

Increase

Diagnosis and Prognosis

[131]

6

miR-23a

Serum

Decrease

Diagnosis and Prognosis

[132]

7

miR-150-5p, miR-142-3p

Serum

Decrease

Diagnosis and Prognosis

[133]

8

miR-206

Serum

Decrease

Prognosis

[134]

9

miR-138

WB

Decrease

Diagnosis and Prognosis

[135]

10

let-7g-5p

Serum

Decrease

Prognosis

[136]

11

miR-495-3p miR-376c-3p miR-6730-3p

melanoma cell

Increase

Prognosis

[137]

12

miR-17, miR-19a, miR-21, miR-126, miR-149

melanoma cell

Increase

Prognosis

[138]

13

miR-532-5p miR-106b

melanoma cell

Increase

Prognosis

[139]

14

miR-125b

Serum

Decrease

Prognosis

[140]

15

miR-1180-3p

Serum

Decrease

Diagnosis and Prognosis

[141]

16

miR-1246, miR-185

Plasma

Increase

Prognosis

[142]

Leukemia

1

miR-10b

Blood

Increase

Prognosis

[143]

2

miR-125b

Plasma

Increase

Prognosis

  [144]

3

miR-21

BM

Increase

Diagnosis and Prognosis

  [145]

4

miR-23b-5p

Blood

Decrease

Diagnosis

[146]

5

miR-155-5p

Serum

Increase

Diagnosis

[147]

6

miR-181b-5p

Plasma

Decrease

Diagnosis

 [148]

7

miR-320d

Serum

Increase

Diagnosis and Prognosis

 [149]

Lymphoma

1

miR-22

Serum

Increase

Diagnosis and Prognosis

[150]

2

miR-485-3p

Plasma

Increase

Diagnosis

[151]

3

miR-375-3p, miR-107

Plasma

Decrease

Diagnosis

[151]

4

miR‐483‐3p and miR‐451a

Serum

Decrease

Diagnosis and Prognosis

[152]

5

miR‐379‐5p, miR‐135a‐3p and miR‐4476

Serum

Increase

Diagnosis and Prognosis

[152]

6

miR-3960, miR-6089, miR-939-5p

Plasma

Decrease

Diagnosis

[153]

7

miR-124, miR-532-5p

Plasma

Increase

Diagnosis

[154]

8

miR-425, miR-145

Plasma

Decrease

Diagnosis

[154]

9

miR-21-5p, miR-15a-3p

Serum

Increase

Diagnosis

[155]

10

miR-181a-5p, miR-210-5p

Serum

Decrease

Diagnosis

[155]

11

miR-125b-5p, miR-99a-5p

Serum

Increase

Prognosis

[156]

miR: Micro ribonucleic acid.

  1. Lines 1125 to 1134 is a duplication of lines 1114 to 1124.
  • Thank you for pointing this out. We corrected the duplicated section.
  1. In line 872, the authors mentioned the …the levels of these miRNAs in exosomes. Which miRNAs?? Also provide reference.
  • Thank you for pointing this out. We changed the whole section as follows:
    • Exosomes extracted from various bodily fluids and tissue samples are examined as potential sources of biomarkers for the diagnosis and prognosis of ovarian cancer. Serum-derived exosomes from patients with ovarian cancer had higher levels of miR-200c, miR-145, and miR-93 than those from patients with benign illness and borderline ovarian cancer, according to an analysis of a collection of miRNAs that are overexpressed in ovarian cancer. Compared to stage I–II patients, stage III–IV individuals (which involves those with lymph node metastases) have significantly greater expression of miR-200b and miR-200c. Exosomes from patients with epithelial ovarian cancer exhibit greater levels of miR-146b-5p expression than those from the healthy control group. Because they are readily available samples, exosomes in urine have recently caught the interest of researchers. Patients with ovarian cancer reported greater levels of miR-92a expression in urine-derived exosomes.
    • urine-derived exosomes.
  1. Table 3A-…Saliva not Salvia.
  • As recommended, the wrong word was corrected.

  1. There were many points that the authors could have used simple phrases. Used many words to explain a simple point. The review could be compressed into half of the pages used.
  • Thank you for pointing this out. We tried to provide a comprehensive overview; however, as you suggested, we removed the wordy sections and re-write some sections and tried to remove some parts from the manuscripts.

Reviewer 2 Report

Comments and Suggestions for Authors

This paper is a review on the clinical potential of exosomes as biomarkers for liquid biopsia. This approach is interesting, and different from other reviews. However, I think the manuscript is over ambitious and raises some concerns:

  1. More than half of the references are dated before 2020. This is a fast moving field, and this type of paper should be more updated. The information about initial studies of biomarkers that have not been implemented after more than 10 years should be handled carefully.
  2. The manuscript has not taken into account the recommendation of the International Society for Extracellular Vesicles (ISEV). It is advised to use the term Small Extracellular Vesicles, since with the actual isolation routes it is not possible to differentiate origin of the vesicles. doi: 10.1002/jev2.12404.
  3. Section 3, Lines 170-171. “Exosomes are extremely stable….” I do not agree with that sentence.  I think it is not well supported. The storing conditions are critical.
  4. The attempt to revise the techniques for characterization of exosomes is beyond the scope of the paper.
  5. The information about clinical trials in interesting and original. I think the manuscript should be rewritten foccussing in that part, and leaving empty parragraphs with no conclusions about old studies about exosomal biomarkers that have not been supported further.

Author Response

We would like to thank you for the opportunity to resubmit a revised copy of the manuscript biomolecules-3512362 entitled “Exosomal Biomarkers: A Comprehensive Overview of Diagnostic and Prognostic Applications in Malignant and Non-Malignant Disorders”. We also appreciate the positive feedback and helpful comments of the reviewers. The manuscript has been revised to address the comments, which are appended alongside our responses to this letter.

Sincerely yours,

Davood Bashash, Ph.D.

Associate Professor of Hematology

Reviewer #2:

This paper is a review on the clinical potential of exosomes as biomarkers for liquid biopsia. This approach is interesting, and different from other reviews. However, I think the manuscript is over ambitious and raises some concerns:

  • Thank you for your kind and positive feedback. We truly appreciate your time and help.

  1. More than half of the references are dated before 2020. This is a fast moving field, and this type of paper should be more updated. The information about initial studies of biomarkers that have not been implemented after more than 10 years should be handled carefully.
  • Thank you for pointing this out. We rechecked all the cited references and tried to either provide new references or change the whole data at those dated sections like:
    • Opportunities biomarkers for the diagnosis of asthma are exosomal miRNAs. These miRNAs have different expression patterns in asthmatic patients than in healthy people, and their levels are correlated with the phenotype and severity of the condition. To distinguish individuals with moderate non-symptomatic asthma from healthy populations, for example, the exosomal miRNAs from the BALF of asthmatic patients vary significantly from those of healthy control participants. Regarding the low type-2 asthma linked to obesity, it has been shown that plasma exosomal miRNA signatures contribute to lung function decline, which forms the foundation for a stratified treatment response. In addition, there are other phenotypes of asthma, such as severe asthma, non-allergic asthma, and allergic asthma. Differentiating between distinct phenotypes with the use of exosomal miRNA profiles enables more individualized treatment strategies. For instance, whilst lower levels of miR-21-5p, miR-126-3p, and miR-146a-5p may indicate neutrophilic asthma, higher exosomal levels of these two markers are suggestive of T2high atopic asthma.

As bladder cancer is a diverse illness marked by a high mutation load, multiple studies emphasized the need of integrated molecular patterns better than single gene testing which may be faulty in certain tumors but not others. Huang et al. used RNA sequencing to find an RNA panel made up of two lncRNAs (MIR205HG and GAS5) and three mRNAs (KLHDC7B, CASP14, and PRSS1) that can differentiate bladder cancer patients from healthy participants (AUC=0.924, 95% CI, 0.875–0.974). The amounts of expression of these five RNAs were also linked to clinicopathological characteristics. The amounts of expression of an exosome lncRNA panel (UCA1-201, UCA1-203, MALAT1, and LINC00355) were also shown to have excellent sensitivity and specificity in distinguishing urothelial cancer from healthy tissue (92% sensitivity and 91.7% specificity) in a research by Yazarlou et al. (n=108).

Zhang et al. found that serum samples from patients with clear cell RCC had higher levels of exosomal miR-210 and miR-1233 than healthy controls, and that these levels decreased following nephrectomy. The authors came to the conclusion that exosomal miR-210 and miR-1233 could be useful markers for low-invasive diagnosis and follow-up with patients with clear cell RCC.

Exosomes extracted from various bodily fluids and tissue samples are examined as potential sources of biomarkers for the diagnosis and prognosis of ovarian cancer. Serum-derived exosomes from patients with ovarian cancer had higher levels of miR-200c, miR-145, and miR-93 than those from patients with benign illness and borderline ovarian cancer, according to an analysis of a collection of miRNAs that are overexpressed in ovarian cancer. Compared to stage I–II patients, stage III–IV individuals (which involves those with lymph node metastases) have significantly greater expression of miR-200b and miR-200c. Exosomes from patients with epithelial ovarian cancer exhibit greater levels of miR-146b-5p expression than those from the healthy control group. Because they are readily available samples, exosomes in urine have recently caught the interest of researchers. Patients with ovarian cancer reported greater levels of miR-92a expression in urine-derived exosomes.

  1. The manuscript has not taken into account the recommendation of the International Society for Extracellular Vesicles (ISEV). It is advised to use the term Small Extracellular Vesicles, since with the actual isolation routes it is not possible to differentiate origin of the vesicles. doi: 10.1002/jev2.12404.
  • Thank you for pointing this out. We understand that exosome defines a subtype of small EVs (mentioned in “doi: 10.1002/jev2.12404”). The reason why we used the term “exosome” was that we used the original studies whose main target was exosomes and indeed we reflected their results. Therefore, we utilized “exosome” instead of extracellular vesicle. We tried to reflect this fact in the manuscript as follows:
    • Exosome is a biogenesis‐related term exhibiting origin from the endosomal system. Being typically smaller than 200 nm, exosomes are a subtype of small extracellular vesicles (EVs). Indeed, they were formerly thought to be extracellular vesicles that expel undesirable cellular trash.
  1. Section 3, Lines 170-171. “Exosomes are extremely stable….” I do not agree with that sentence. I think it is not well supported. The storing conditions are critical.
  • As recommended, we tried to provide supporting information for that section as follows:
    • Exosomes have slower rate of degradation and are extremely stable in stored patient samples. Indeed, as exosomes are membranous structures, the contents could be adequately protected from degradation by extracellular proteases and can be highly stable in storage conditions. This makes the application of exosomes in clinical settings possible because samples may be stored for extended periods of time before being analyzed. Due to the fact that exosomes exhibit surface markers that are associated with their cellular origin, it is possible to trace them back to their point of origin. Nonetheless, the stability of exosomes could be influenced by storing conditions. In this regard, the biodistribution and uptake efficiency could be notably reduced following storage at 4 °C and −20 °C. Exosomes, however, maintain their biodistribution at −80 °C for 14 days. Also, it was shown that the number of exosomes, their size, their capacity to express protein at the surface, and functional stability could be constant at −70 °C after 25 days.
  1. The attempt to revise the techniques for characterization of exosomes is beyond the scope of the paper.
  • Thank you for the suggestion. We understand that focusing on technical features could distract the readers from the main concept, and we value your comment. We inserted the data related to techniques under the title “3.3. Limitations and overcoming limitations of Exosome diagnostic use in the clinic”, and we tried to remove some parts to integrate the section and not focus on technical features.
  1. The information about clinical trials in interesting and original. I think the manuscript should be rewritten foccussing in that part, and leaving empty parragraphs with no conclusions about old studies about exosomal biomarkers that have not been supported further.
  • Thanks a lot for the positive feedback. Indeed, we tried to investigate the situation of clinical trials in the context of diagnostic and prognostic features of exosomes. Although we conducted an analysis of clinical trials, the study was not in accordance with the systematic analysis guidelines; therefore, we designed the manuscript based on a narrative review article. In order to support the main concept of using exosomes in clinical trials, we tried to provide other relevant studies (mainly clinical ones) to enrich all aspects of the main title, and that was why we integrated two separated viewpoints into one review article.

Reviewer 3 Report

Comments and Suggestions for Authors

Exosomal Biomarkers: A Comprehensive Overview of Diagnos-2 tic and Prognostic Applications in Malignant and Non-Malignant Disorders. By  Mahda Delshad et al

Of course the use of exosomes as a source of biomarkers for a broad panel of human diseases is a warm issue. However, a so ambitious title should include at least all the clinical studies published to date.

In this sense this review misses a list of clinical studies that clearly showed the relevance of exosomes as disease biomarkers in various body fluid. Moreover, the review does not emphasize how much the clinical data did not confirm the relevance of exosomes as a source of new and specific biomarkers; rather the importance of exosomes as a way to render more specific the old biomarkers. The following reference should be included and better commented by the authors:

Fais S, Logozzi M. The Diagnostic and Prognostic Value of Plasmatic Exosome Count in Cancer Patients and in Patients with Other Pathologies. Int J Mol Sci. 2024 Jan 15;25(2):1049. doi: 10.3390/ijms25021049. PMID: 38256122; PMCID: PMC10816819.

Comments on the Quality of English Language

should be improved

Author Response

We would like to thank you for the opportunity to resubmit a revised copy of the manuscript biomolecules-3512362 entitled “Exosomal Biomarkers: A Comprehensive Overview of Diagnostic and Prognostic Applications in Malignant and Non-Malignant Disorders”. We also appreciate the positive feedback and helpful comments of the reviewers. The manuscript has been revised to address the comments, which are appended alongside our responses to this letter.

Sincerely yours,

Davood Bashash, Ph.D.

Associate Professor of Hematology

Reviewer #3:

  1. Of course the use of exosomes as a source of biomarkers for a broad panel of human diseases is a warm issue. However, a so ambitious title should include at least all the clinical studies published to date.
  • Thanks a lot for your positive feedback. Indeed, we tried to gather information and digest useful data in order to provide a narrative review article. Although we conducted an analysis of clinical trials, the study was not in accordance with the systematic analysis guidelines; nevertheless, we believe that we covered all clinical trials listed on clinicaltrials.gov based on the mentioned inclusion and exclusion criteria. Furthermore, we tried to support the main idea of using exosomes in clinical trials by providing published clinical studies in the previous sections. Having said that, we do not claim that we could cover all clinical studies, but we tried our best.

  1. In this sense this review misses a list of clinical studies that clearly showed the relevance of exosomes as disease biomarkers in various body fluid. Moreover, the review does not emphasize how much the clinical data did not confirm the relevance of exosomes as a source of new and specific biomarkers; rather the importance of exosomes as a way to render more specific the old biomarkers. The following reference should be included and better commented by the authors: Fais S, Logozzi M. The Diagnostic and Prognostic Value of Plasmatic Exosome Count in Cancer Patients and in Patients with Other Pathologies. Int J Mol Sci. 2024 Jan 15;25(2):1049. doi: 10.3390/ijms25021049. PMID: 38256122; PMCID: PMC10816819.
  • Thank you for pointing this out. We tried to cover the clinical studies whose main objectives were related to using exosomes in body fluid. For instances, Ref 13 (10.7150/jca.48531) used serum, Ref 127 (10.1016/j.jalz.2014.06.008), and Ref 138 (10.1007/s13365-015-0325-3) used blood and urine. As your valuable comment could enrich the manuscript, we tried to focus more on body fluid samples in a new section as follows:
    • A sequence of imaging scans is used to diagnose different tumor forms, and a biopsy is then performed to verify the diagnosis. In addition to being intrusive, these techniques may be expensive and painful for the patient. Additionally, individuals are more probable to acquire a late-stage diagnosis if regular screening is not performed. One possible disadvantage of a delayed diagnosis is that it may reduce the patient's reaction to treatment and, therefore, their chances of survival. Because of this, it's critical to track how well a patient is responding to treatment. Many of traditional cancer detection methods are inappropriate for patients because they need intrusive procedures, such as surgeries, to collect tissue samples for examination. These operations may be risky and uncomfortable. Furthermore, they may be expensive, burdening patients and healthcare systems alike, especially when many tests are needed to verify a diagnosis or track the effectiveness of therapy. As a consequence, smaller tumors or malignant cells may go undetected until they have progressed to more severe and un-treatable phases. A growing effort has been made to develop less invasive, cost-effective, highly sensitive diagnostic techniques, including liquid biopsies and sophisticated imaging technologies, that have the potential to detect tumors early and monitor patients undergoing therapy in order to conquer these limitations. By providing earlier detection, less invasiveness, and more accurate monitoring, these advancements have the potential to completely transform the diagnosis of cancer and strengthen our fight against this terrible illness.

In the area of cancer detection, exosomes have become a rapidly developing research area with enormous promise. Exosomes are also essential for a number of pathological as well as physiological processes such as immunological responses, cancer, pregnancy issues, and cardiovascular problems. Exosomes were discovered to be present in almost every bodily fluid, including blood, urine, Cerebrospinal fluid (CSF), saliva, pleural effusion, ascites fluid, amniotic fluid, breast milk, and bronchoalveolar lavage fluid (BALF), according to the many investigations that followed. The pilot study evaluated saliva collection methods for exosome isolation and characterized exosomal proteins in cancer-free controls, oral potentially malignant disease (OPMD), and oral squamous cell carcinoma (OSCC) patients. A three-protein panel (AMER3, LOXL2, AL9A1) distinguished cancer-free individuals from OPMD/OSCC with high accuracy (AUC 0.93). Another panel (PSB7, AMER3, LOXL2) showed strong diagnostic potential for OSCC (AUC 0.96), suggesting its promise as a biomarker for early detection. Another study found significant differences in the size, concentration, and biomolecular composition of salivary exosomes between healthy individuals, tobacco consumers, and oral cancer patients. Tobacco users and cancer patients had larger and more concentrated exosomes, suggesting early cellular changes. These findings highlight the potential of salivary exosomes as noninvasive biomarkers for distinguishing high-risk individuals and enabling early oral cancer detection. Sweat is a simple and less interference-prone biofluid, making it valuable for disease biomarker studies. Sweat exosome-based detection shows potential for lung cancer screening through molecular expression analysis. Notably, Dermcidin (DCD) and Prolactin Inducible Protein (PIP) serve as biomarkers, with DCD linked to breast cancer and lymph nodes, while PIP is overexpressed in breast and prostate cancer. Tear exosomes contain higher levels of exosome markers (CD9, CD63) than serum exosomes and were analyzed using qRT-PCR and western blot. Breast cancer-specific miR-21 and miR-200c were highly expressed in tear exosomes from meta-static breast cancer patients compared to healthy individuals. Cholangiocarcinoma (CCA) is an aggressive cancer with poor prognosis and limited diagnostic or treatment options. This study identified two exosomal lncRNAs, ENST00000588480.1 and ENST00000517758.1, with significantly increased expression in bile from CCA patients. Their combined diagnostic performance (AUC: 0.709, sensitivity: 82.9%, specificity: 58.9%) correlated with cancer progression and poor survival. KEGG analysis highlighted their involvement in the p53 signaling pathway, suggesting their potential as non-invasive bile-based biomarkers and therapeutic targets for CCA.

  • Furthermore, exosomes are a new source of biomarkers capable of serving as diagnostic and prognostic tools. As they are stable, they could maintain the cargoes related to a specific type of disease or a pathologic disorder more constant and that is why they are novel potent specific biomarkers. Moreover, their count could correlate with the burden of disorders such as cancers; therefore, exosome quantification can be a non-invasive method for cancer monitoring and prognosis (10.1016/j.jalz.2014.06.008). We inserted a new section as follows:
    • We demonstrated that exosomes could serve as diagnostic and prognostic biomarkers, particularly in cancer, where their unique molecular cargo provides insights into tumor progression. Indeed, exosomes are a new source of biomarkers capable of serving as diagnostic and prognostic tools. As they are stable, they could maintain the cargoes related to a specific type of disease or a pathologic disorder more constant and that is why they are potent specific biomarkers. Furthermore, their count could correlate with the burden of disorders such as cancers; therefore, exosome quantification can be a non-invasive method for cancer monitoring and prognosis.
  1. Comments on the Quality of English Language: should be improved
  • Thank you for pointing this out. We tried to integrate the manuscript and re-check all sections and corrected the misunderstandings and errors.

Round 2

Reviewer 1 Report

Comments and Suggestions for Authors

The authors addressed all my original concerns.  In my opinion the information in this review is significant and should be made available to the scientific community.

Author Response

Reviewer #1:

The authors addressed all my original concerns.  In my opinion the information in this review is significant and should be made available to the scientific community.

Thank you for your kind and positive feedback. We are sure that your insightful comments and thoughtful insights greatly helped enrich our manuscript.

Reviewer 3 Report

Comments and Suggestions for Authors

Exosomal Biomarkers: A Comprehensive Overview of Diagnos-2 tic and Prognostic Applications in Malignant and Non-Malignant Disorders. By  Mahda Delshad et al

This reviewer apologies for not agreeing on what the authors actually replied to the previous report.

In particular the authors write: “Thanks a lot for your positive feedback. Indeed, we tried to gather information and digest useful data in order to provide a narrative review article. Although we conducted an analysis of clinical trials, the study was not in accordance with the systematic analysis guidelines; nevertheless, we believe that we covered all clinical trials listed on clinicaltrials.gov based on the mentioned inclusion and exclusion criteria. Furthermore, we tried to support the main idea of using exosomes in clinical trials by providing published clinical studies in the previous sections. Having said that, we do not claim that we could cover all clinical studies, but we tried our best”

This reviewer is very much disappointed for this reply for the following reasons:

  1. If literature on clinical trials should be based on exclusively those trials reported on clinicaltrials.gov probably many interesting studies did not contribute to increase the current knowledge
  2. The expectation that exosomes contain new tumor biomarkers has been disappointed actually; inasmuch as the data were and are continuing to be very delusive and confusing. While it is clear that the number of circulating exosomes is always higher in tumor patients than in controls and decrease after surgical treatment; in turn meaning that the increased exosomes’burden is due to the tumor release. For this reason it has been proposed to include the count of plasmatic exosomes into the blood count as a new and valuable new tool of both diagnostic and prognostic value (Logozzi M, Orefice NS, Di Raimo R, Mizzoni D, Fais S. The Importance of Detecting, Quantifying, and Characterizing Exosomes as a New Diagnostic/Prognostic Approach for Tumor Patients. Cancers (Basel). 2023 May 23;15(11):2878. doi: 10.3390/cancers15112878. PMID: 37296842; PMCID: PMC10251946.). This issue deserves new discussion and all the suggested references should be included:
  • Logozzi M, De Milito A, Lugini L, Borghi M, Calabrò L, Spada M, Perdicchio M, Marino ML, Federici C, Iessi E, Brambilla D, Venturi G, Lozupone F, Santinami M, Huber V, Maio M, Rivoltini L, Fais S. High levels of exosomes expressing CD63 and caveolin-1 in plasma of melanoma patients. PLoS One. 2009;4(4):e5219. doi: 10.1371/journal.pone.0005219. Epub 2009 Apr 17. PMID: 19381331; PMCID: PMC2667632.
  • Logozzi M, Angelini DF, Giuliani A, Mizzoni D, Di Raimo R, Maggi M, Gentilucci A, Marzio V, Salciccia S, Borsellino G, Battistini L, Sciarra A, Fais S. Increased Plasmatic Levels of PSA-Expressing Exosomes Distinguish Prostate Cancer Patients from Benign Prostatic Hyperplasia: A Prospective Study. Cancers (Basel). 2019 Sep 27;11(10):1449. doi: 10.3390/cancers11101449. PMID: 31569672; PMCID: PMC6826376.
  • Logozzi M, Mizzoni D, Capasso C, Del Prete S, Di Raimo R, Falchi M, Angelini DF, Sciarra A, Maggi M, Supuran CT, Fais S. Plasmatic exosomes from prostate cancer patients show increased carbonic anhydrase IX expression and activity and low pH. J Enzyme Inhib Med Chem. 2020 Dec;35(1):280-288. doi: 10.1080/14756366.2019.1697249. PMID: 31790614; PMCID: PMC6896418.
  • Logozzi M, Mizzoni D, Di Raimo R, Giuliani A, Maggi M, Sciarra A, Fais S. Plasmatic Exosome Number and Size Distinguish Prostate Cancer Patients From Healthy Individuals: A Prospective Clinical Study. Front Oncol. 2021 Oct 20;11:727317. doi: 10.3389/fonc.2021.727317. PMID: 34745949; PMCID: PMC8564386.
  • Rodríguez-Zorrilla S, Lorenzo-Pouso AI, Fais S, Logozzi MA, Mizzoni D, Di Raimo R, Giuliani A, García-García A, Pérez-Jardón A, Ortega KL, Martínez-González Á, Pérez-Sayáns M. Increased Plasmatic Levels of Exosomes Are Significantly Related to Relapse Rate in Patients with Oral Squamous Cell Carcinoma: A Cohort Study. Cancers (Basel). 2023 Dec 2;15(23):5693. doi: 10.3390/cancers15235693. PMID: 38067397; PMCID: PMC10705147.

  1. The acidic microenvironment has a key role in the increase of the exosome release by tumor cells and a key paper has been published in this sense:

Logozzi M, Mizzoni D, Angelini DF, Di Raimo R, Falchi M, Battistini L, Fais S. Microenvironmental pH and Exosome Levels Interplay in Human Cancer Cell Lines of Different Histotypes. Cancers (Basel). 2018 Oct 5;10(10):370. doi: 10.3390/cancers10100370. PMID: 30301144; PMCID: PMC6210604.

The endorsement of this reviewer passes through a real reappraisal of this issue in a new revised version

Author Response

Reviewer #3:

This reviewer is very much disappointed for this reply for the following reasons:

  1. If literature on clinical trials should be based on exclusively those trials reported on clinicaltrials.gov probably many interesting studies did not contribute to increase the current knowledge
  • Thank you for the recommendation. Given that clinicaltrials.gov is the most comprehensive website for clinical trials, we tried to collect all available trials up to now. As we have already mentioned, this manuscript is a narrative article, and we tried to use around 280 references to cover all aspects. It is worth noting that, following your kind comment, we added articles that you recommended to our manuscript.
  • The expectation that exosomes contain new tumor biomarkers has been disappointed actually; inasmuch as the data were and are continuing to be very delusive and confusing. While it is clear that the number of circulating exosomes is always higher in tumor patients than in controls and decrease after surgical treatment; in turn meaning that the increased exosomes’burden is due to the tumor release. For this reason it has been proposed to include the count of plasmatic exosomes into the blood count as a new and valuable new tool of both diagnostic and prognostic value (Logozzi M, Orefice NS, Di Raimo R, Mizzoni D, Fais S. The Importance of Detecting, Quantifying, and Characterizing Exosomes as a New Diagnostic/Prognostic Approach for Tumor Patients. Cancers (Basel). 2023 May 23;15(11):2878. doi: 10.3390/cancers15112878. PMID: 37296842; PMCID: PMC10251946.). This issue deserves new discussion and all the suggested references should be included: Logozzi M, De Milito A, Lugini L, Borghi M, Calabrò L, Spada M, Perdicchio M, Marino ML, Federici C, Iessi E, Brambilla D, Venturi G, Lozupone F, Santinami M, Huber V, Maio M, Rivoltini L, Fais S. High levels of exosomes expressing CD63 and caveolin-1 in plasma of melanoma patients. PLoS One. 2009;4(4):e5219. doi: 10.1371/journal.pone.0005219. Epub 2009 Apr 17. PMID: 19381331; PMCID: PMC2667632. Logozzi M, Angelini DF, Giuliani A, Mizzoni D, Di Raimo R, Maggi M, Gentilucci A, Marzio V, Salciccia S, Borsellino G, Battistini L, Sciarra A, Fais S. Increased Plasmatic Levels of PSA-Expressing Exosomes Distinguish Prostate Cancer Patients from Benign Prostatic Hyperplasia: A Prospective Study. Cancers (Basel). 2019 Sep 27;11(10):1449. doi: 10.3390/cancers11101449. PMID: 31569672; PMCID: PMC6826376. Logozzi M, Mizzoni D, Capasso C, Del Prete S, Di Raimo R, Falchi M, Angelini DF, Sciarra A, Maggi M, Supuran CT, Fais S. Plasmatic exosomes from prostate cancer patients show increased carbonic anhydrase IX expression and activity and low pH. J Enzyme Inhib Med Chem. 2020 Dec;35(1):280-288. doi: 10.1080/14756366.2019.1697249. PMID: 31790614; PMCID: PMC6896418. Logozzi M, Mizzoni D, Di Raimo R, Giuliani A, Maggi M, Sciarra A, Fais S. Plasmatic Exosome Number and Size Distinguish Prostate Cancer Patients From Healthy Individuals: A Prospective Clinical Study. Front Oncol. 2021 Oct 20;11:727317. doi: 10.3389/fonc.2021.727317. PMID: 34745949; PMCID: PMC8564386. Rodríguez-Zorrilla S, Lorenzo-Pouso AI, Fais S, Logozzi MA, Mizzoni D, Di Raimo R, Giuliani A, García-García A, Pérez-Jardón A, Ortega KL, Martínez-González Á, Pérez-Sayáns M. Increased Plasmatic Levels of Exosomes Are Significantly Related to Relapse Rate in Patients with Oral Squamous Cell Carcinoma: A Cohort Study. Cancers (Basel). 2023 Dec 2;15(23):5693. doi: 10.3390/cancers15235693. PMID: 38067397; PMCID: PMC10705147.
  • Thank you for your kind suggestion. We value your suggestion; therefore, we added a section to express this idea and discuss it elaborately, as follows:
    • Furthermore, their count could correlate with the burden of disorders such as cancers; therefore, exosome quantification can be a non-invasive method for cancer monitoring and prognosis. Studies have emphasized the diagnostic and prognostic value of quantifying plasmatic exosome levels in cancer patients. Indeed, the exosome levels are significantly higher in tumor patients and decrease after tumor removal. The exosome count could be a potential addition to routine blood tests, as it could be measured easily. Standardizing exosome detection and quantification methods could be one of the future areas in diagnostic and prognostic studies. All in all, it could be inferred that the count of exosomes or tumor antigen-expressing ones could be another advantage of exosomes in the diagnostic and prognostic assessments as they can be effortlessly measured and are associated with the development of various tumors such as melanoma and prostate cancer.

  • Also, the mentioned articles were added in the relevant section, with emphasis on the level and/or the size of exosomes as follows:
    • Additionally, Logozzi et al. provided a novel non-invasive method that enables the identification and measurement of human exosome levels in melanoma patients' plasma. According to their findings, the in-house sandwich ELISA (Exotest) for identifying plasma exosomes with tumor-associated antigens could be an innovative clinical management tool for cancer patients.
    • The levels of exosomes could play an important role in providing the diagnostic and prognostic overview of prostate cancer. Exo-PSA consensus score (EXOMIX), immunocapture-based ELISA (IC-ELISA), and nanoscale flow-cytometry (NSFC) demonstrated 98% to 100% specificity and sensitivity for BPH-PCa discrimination, while statistical analysis revealed that the levels of plasmatic exosomes expressing both CD81 and PSA were significantly higher in PCa compared to both BPH and healthy donors, reaching 100% specificity and sensitivity in differentiating PCa patients from healthy individuals. Similarly, exosomal CA IX expression levels and activity may be used as a biomarker of PCa cancer development. Another study showed that compared to urological disease, PCa had substantially greater plasmatic levels of exosomes, and the exosomes were smaller. The study introduced a non-invasive exosome-based clinical approach for prostate cancer follow-up and early diagnosis, potentially serving as a screening test.
    • Moreover, exosomal plasmatic levels before surgery could be a reliable indicator of survival and early recurrence in OSCC. It was shown that identifying peripheral exosomes can be a new clinical treatment technique for OSCC that may have consequences for prognostic evaluation.
  1. The acidic microenvironment has a key role in the increase of the exosome release by tumor cells and a key paper has been published in this sense: Logozzi M, Mizzoni D, Angelini DF, Di Raimo R, Falchi M, Battistini L, Fais S. Microenvironmental pH and Exosome Levels Interplay in Human Cancer Cell Lines of Different Histotypes. Cancers (Basel). 2018 Oct 5;10(10):370. doi: 10.3390/cancers10100370. PMID: 30301144; PMCID: PMC6210604.
  • We have taken this interesting issue into account and tried to mention the effect of acidic microenvironment. We added the reference as well:
    • For the first time, authors provided proof that tumor acidity and exosome the amount represent common cancer phenotypes. The results of an intriguing study revealed that pH 6.5 caused an astonishing rise in exosome release, and buffering the medium drastically decreased the exosome release in all cancers.